# Transmission of SARS-CoV-2 in free-ranging white-tailed deer in the United States

Aijing Feng [1,2,3], Sarah Bevins [4], Jeff Chandler[5], Thomas J. DeLiberto [6] ✉, Ria Ghai[7], Kristina Lantz[8], Julianna Lenoch[4], Adam Retchless[9], Susan Shriner [5], Cynthia Y. Tang [1,3,10], Suxiang Sue Tong[9], Mia Torchetti[8], Anna Uehara[9] & Xiu-Feng Wan [1,2,3,10,11] ✉

SARS-CoV-2 is a zoonotic virus with documented bi-directional transmission between people and animals. Transmission of SARS-CoV-2 from humans to free-ranging white-tailed deer (*Odocoileus virginianus*) poses a unique public health risk due to the potential for reservoir establishment where variants may persist and evolve. We collected 8,830 respiratory samples from free-ranging white-tailed deer across Washington, D.C. and 26 states in the United States between November 2021 and April 2022. We obtained 391 sequences and identified 34 Pango lineages including the Alpha, Gamma, Delta, and Omicron variants. Evolutionary analyses showed these white-tailed deer viruses originated from at least 109 independent spillovers from humans, which resulted in 39 cases of subsequent local deer-to-deer transmission and three cases of potential spillover from white-tailed deer back to humans. Viruses repeatedly adapted to white-tailed deer with recurring amino acid substitutions across spike and other proteins. Overall, our findings suggest that multiple SARS-CoV-2 lineages were introduced, became enzootic, and co-circulated in white-tailed deer.

Severe acute respiratory disease syndrome coronavirus-2 (SARS-CoV-2) is a zoonotic virus[1] similar to other high-consequence coronaviruses, including severe acute respiratory syndrome coronavirus and Middle East respiratory syndrome coronavirus[2]. Since its emergence in 2019, SARS-CoV-2 has evolved rapidly and produced numerous SARS-CoV-2 genetic variants, including Variants of Concern (VOCs) Alpha, Beta, Gamma, Delta, and Omicron[3]. In addition to humans, SARS-CoV-2 infections have been documented in a wide range of wild, domestic, and exotic animals in captivity, such as deer[4], mink[5–7], rats[8], otters, ferrets, hamsters, gorillas, cats, dogs, lions, and tigers[9]. Further, SARS-CoV-2 transmission from animals to humans, while not common, has been documented or suspected in farmed mink (*Neogale vison*)[5, 6], domestic cats (*Felis catus*)[10], and white-tailed deer (*Odocoileus virginianus*)[11], highlighting animals as potential reservoirs for secondary zoonotic infections. An animal reservoir for SARS-CoV-2 refers to a host in which the virus circulates covertly, persisting in the population and can be transmitted to other animals or humans potentially causing disease outbreaks.

[1]Center for Influenza and Emerging Infectious Diseases, University of Missouri, Columbia, MO, USA. [2]Department of Molecular Microbiology and Immunology, School of Medicine, University of Missouri, Columbia, MO, USA. [3]Bond Life Sciences Center, University of Missouri, Columbia, MO, USA. [4]USDA APHIS Wildlife Services National Wildlife Disease Program, Fort Collins, CO, USA. [5]National Wildlife Research Center, Wildlife Services, Animal and Plant Health Inspection Service, US Department of Agriculture, Fort Collins, CO, USA. [6]USDA APHIS Wildlife Services, Fort Collins, CO, USA. [7]One Health Office, National Center for Emerging and Zoonotic Infectious Diseases, Centers for Disease Control and Prevention, Atlanta, GA, USA. [8]National Veterinary Services Laboratories, Animal and Plant Health Inspection Service, United States Department of Agriculture, Ames, IA, USA. [9]National Center for Immunization and Respiratory Diseases, Centers for Disease Control and Prevention, Atlanta, GA, USA. [10]MU Institute for Data Science and Informatics, University of Missouri, Columbia, MO, USA. [11]Department of Electrical Engineering & Computer Science, College of Engineering, University of Missouri, Columbia, MO, USA. ✉ e-mail: thomas.j.deLiberto@usda.gov; wanx@missouri.edu

White-tailed deer are common in both urban and rural areas in North America with an estimated population of 30 million distributed throughout the United States (US). Damas et al. (2020) showed a high degree of sequence identity between human and white-tailed deer angiotensin converting enzyme 2 (ACE2) proteins[12], and experimental infection studies demonstrated that the (Wuhan-Hu-1 strain)-like SARS-CoV-2 virus can readily infect white-tailed deer and lead to high loads of viral shedding and onward spread to naïve conspecifics[13–15]. Chandler et al. estimated that 40% of tested white-tailed deer were exposed to SARS-CoV-2, starting as early as January 2020 in four states in the US[16]. Subsequently, active SARS-CoV-2 infections, as evidenced by reverse transcription polymerase chain reaction (RT-PCR) detection, were reported in white-tailed deer in the US (i.e., Ohio[4], Iowa[17], Pennsylvania[18], New York[19]) and in Ontario, Canada[11]. The viruses reported to date in white-tailed deer are genetically diverse, including Pango lineages[20] B.1.2 and B.1.311 in Iowa (sampling period, April 2020 to January 2021)[17], B.1.2, B.1.582, B.1.596 in Ohio (January to March 2021)[4], B.1.1.7 (Alpha), AY.88 (Delta), AY.5 (Delta), and AY.103 (Delta) in Pennsylvania (January to November 2021)[18], B.1, B.1.1, B.1.2, B.1.243, B.1.409, B.1.507, B.1.517, B.1.1.7 (Alpha), B.1.1.28 (Gamma), P.1 (Gamma), and B.1.617.2 (Delta) in New York (September 2020 to December 2021)[19], and B.1.641 (Dec 2021) in Ontario (November to December 2021)[11]. Of interest, the majority of these white-tailed deer viruses were genetically related to those that were concurrently circulating in humans. Identification of genetically highly similar viruses from multiple animals captured on two different days in the same or nearby location suggested that SARS-CoV-2 was likely transmitted within white-tailed deer populations[4, 19]. Epidemiological evidence for the possible transmission of SARS-CoV-2 from white-tailed deer to people in Canada has been reported[11].

In this work, we present the results of a large-scale surveillance for SARS-CoV-2 across free-ranging white-tailed deer populations in the US. Our objectives were to understand the genetic diversity of SARS-CoV-2 in free-ranging white-tailed deer, to evaluate whether the virus circulated within white-tailed deer populations, and to assess transmission frequencies associated with zoonotic infections. In total, we collected 8,830 white-tailed deer nasal or oral swabs in Washington, D.C. and 26 states in the US from late fall of 2021 to early spring of 2022, yielding 944 RT-PCR positive samples. Of those, 391 were sequenced and analyzed using molecular and evolutionary approaches. We detected frequent introductions of multiple SARS-CoV-2 lineages in white-tailed deer, which subsequently became enzootic and co-circulated, and certain lineages persisted in white-tailed deer even after their decline in human populations. In addition, three cases of potential spillover from white-tailed deer back to humans were identified. These findings suggest that white-tailed deer could potentially serve as a reservoir for SARS-CoV-2, presenting zoonotic risks to humans. In the context of this manuscript, the term "deer" specifically refers to white-tailed deer.

## Results

### Detecting multiple genetic variants in free-ranging white-tailed deer across the United States

From November 4, 2021 to April 4, 2022, a total of 8830 white-tailed deer oral or nasal samples were collected from Washington, D.C. and 26 states in the US which participated in this study. SARS-CoV-2 was detected in 944 samples by quantitative reverse transcription polymerase chain reaction (qRT-PCR) (Fig. 1a). We sequenced viral genomes from 391 samples with high quality RNA. Among these sequences, 383 from 23 states (Fig. 1b) with complete metadata were used in this study, and 346 of them had a genomic coverage of >50% (Supplementary Data 1). Overall, 282 samples had high sequencing coverage (i.e., >95% of the reference genome) and were selected for further evolutionary analyses.

Pango lineage assignment identified 34 lineages, which belong to B.1 and four VOCs, Alpha ($n = 70$), Gamma ($n = 9$), Delta ($n = 273$), and Omicron ($n = 2$) (Supplementary Data 1). These Pango lineages were widely distributed across the sampled states (Fig. 1c). The Northeast US, including Massachusetts (MA), New York (NY), New Jersey (NJ), Pennsylvania (PA), and West Virginia (WV), had the highest number of Pango lineages. Alpha and Delta variants were detected in white-tailed deer throughout the sampling period, whereas Gamma was only detected early in the sampling period (i.e., November 20 to December 2, 2021), and Omicron only towards the end of our sampling period (i.e., January 24 to February 11, 2022) (Fig. 1d). Throughout the sampling period, Delta and Omicron were predominant in humans, but Alpha and Gamma were infrequently reported.

Taken together, genetically diverse SARS-CoV-2 sub-lineages of Alpha, Gamma, Delta, and Omicron were co-circulating in the white-tailed deer populations across multiple geographic regions of the US. Alpha and Gamma were circulating in the US white-tailed deer populations throughout the study period, although both variants had become rare in humans and were displaced by new variants (Delta and then Omicron).

### Frequent spillovers of SARS-CoV-2 from humans to white-tailed deer

The number of likely independent spillover events from humans to white-tailed deer was determined by identifying a potential precursor virus in humans for each SARS-CoV-2 sequence from white-tailed deer. We identified a potential precursor virus from the GISAID (the Global Initiative on Sharing All Influenza Data) and NCBI (National Center for Biotechnology Information) GenBank databases with the most similar SARS-CoV-2 genomes that was collected in the same state prior to deer sample collection. These potential precursor viruses were integrated with the white-tailed deer SARS-CoV-2 sequences into phylogenetic trees (Supplementary Data 10) and Bayesian phylogenetic analyses performed (Fig. 2a and Supplementary Data 11). The spillover events were determined and grouped into three categories: Human-Deer (at least one precursor human sequence and a single white-tailed deer sequence), Human-Deer-Deer (at least one precursor human sequence and at least two white-tailed deer sequences from multiple individual animals), and Human-Deer-Human (at least one precursor human sequence, at least two white-tailed deer sequences from multiple individual animals, and another human sequence collected after the white-tailed deer sequences) (See Methods).

Out of the 282 white-tailed deer viruses analyzed, 238 were found to be grouped into 109 clusters that also contained human SARS-CoV-2 viruses. For each cluster, a SARS-CoV-2 genomic sequence from a human was identified as the precursor virus with at least 99.85% nucleotide identity, indicating at least one independent spillover event from humans to white-tailed deer (Fig. 2a, Supplementary Data 11, and Supplementary Data 2). In total, 109 independent spillover events were identified, with 106 involving a human SARS-CoV-2 precursor virus from within the same state and the remaining three involving a human SARS-CoV-2 precursor virus that originated from outside the state.

Of these 106 within-state spillover events, 64 were Human-Deer (60 Delta, three Alpha, and one Omicron), 39 were Human-Deer-Deer (29 Delta, eight Alpha, and two Gamma), and 3 were Human-Deer-Human (all Delta) events and were widely distributed across sampled states (Fig. 2a, b). Viruses from NY and NJ had the largest number of Human-Deer-Deer spillover events (Fig. 2d). Nine out of 39 Human-Deer-Deer spillover events contained at least five sequences, with the largest one comprising 17 sequences. The white-tailed deer sequences within each Human-Deer-Deer or Human-Deer-Human spillover event had at least 99.85% sequence identities. Of interest, the white-tailed deer sequences grouped within each Human-Deer-Deer spillover event were sampled from the same county or neighboring counties (Supplementary Fig. 1); of note, all samples in each spillover event were collected within three weeks. The three Human-Deer-Human spillover events included 11, six, and two white-tailed deer sequences each, with

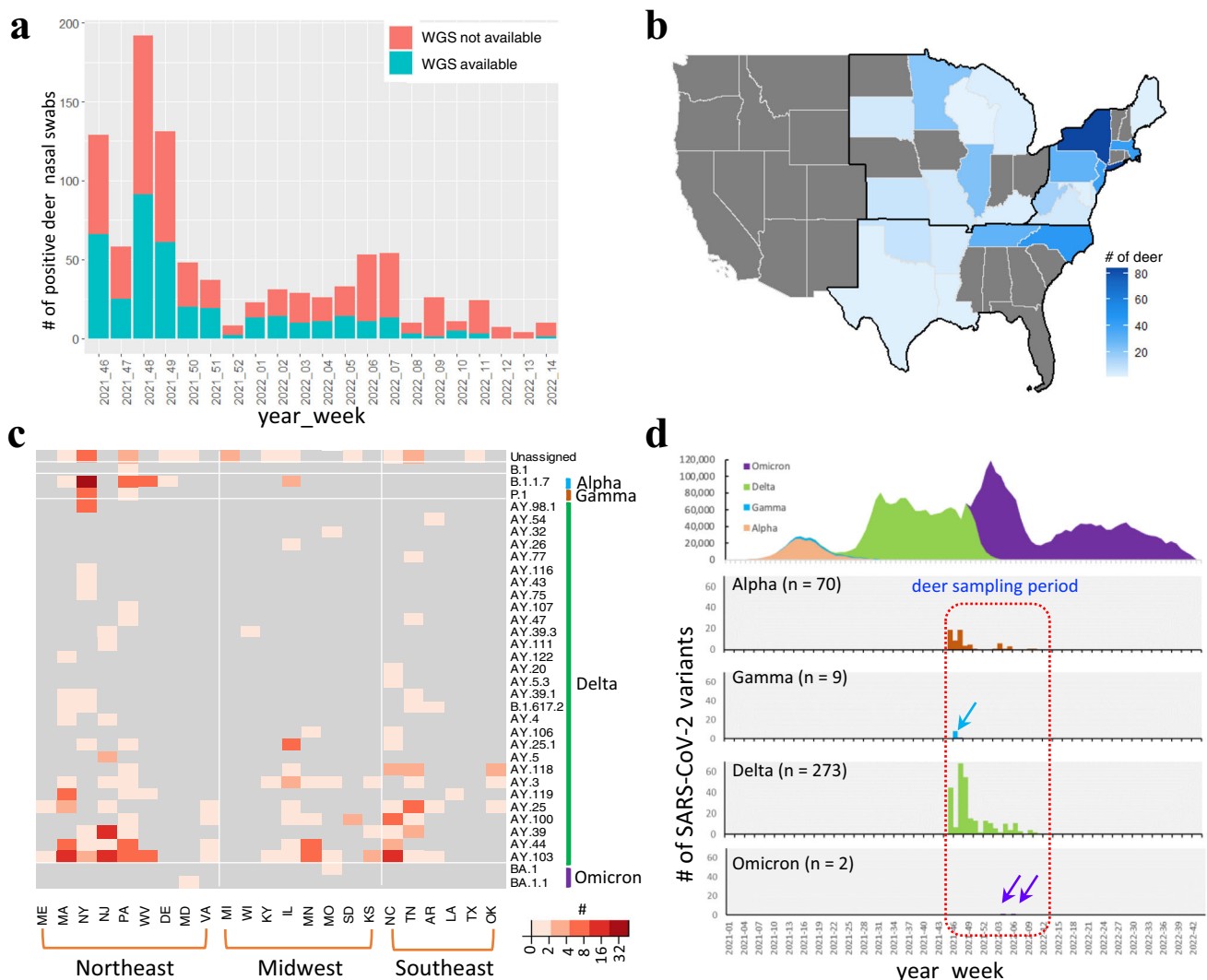

**Fig. 1 | Surveillance of SARS-CoV-2 viruses in free-ranging white-tailed deer in the United States (November 2021-April 2022). a** temporal distribution of SARS-CoV-2 quantitative reverse transcription polymerase chain reaction (qRT-PCR) positive nasal swab samples collected in white-tailed deer and those selected for whole genome sequencing (WGS); **b** geographic distribution of SARS-CoV-2 positive white-tailed deer samples with WGS; **c** SARS-COV-2 genetic variants in white-tailed deer nasal swab samples; **d** circulating SARS-CoV-2 Variants of Concern (VOCs) in humans in the United States (Jan 2021-October 2022) and those sampled in white-tailed deer from November 4, 2021 to April 4, 2022. Out of 383 SARS-CoV-2 WGS with complete metadata (Supplementary Data 1), 355 had identified Pango lineages, including 1 non-VOC variants (B.1) and 354 VOCs (70 Alpha, 9 Gamma, 273 Delta, 2 Omicron); 282 high-quality WGS were used further in evolutionary and transmission analyses. Arrow is used to highlight the small number in a particular week. Note that as d) focuses on the VOCs comparison between human and white-tailed deer, a single white-tailed deer sample, which belongs to a non-VOC variant (B.1), was collected on December 1, 2021 in Pennsylvania and was not shown in panel d. The source data for each subpanel is available in the Source Data file.

all white-tailed deer sequences in each spillover event sampled from the same or neighboring counties.

The three other spillover events (Event #107-109 in Supplementary Data 2) involved out-of-state human SARS-CoV-2 sequences. Of these, all were Human-Deer events and belonged to the Delta variant, involving VA (human)- PA (Deer) (AY.25.1), OH (human)-LA (Deer) (AY.119), VA (human)-WV (Deer) (AY.119), respectively (Supplementary Data 11). For our subsequent analyses, we focused only on the 106 spillover events that involved within-state human SARS-CoV-2 sequences.

Overall, these results suggested the SARS-CoV-2 viruses in white-tailed deer originated from at least 109 independent direct or indirect spillover events from humans, with 106 involving within-state human SARS-CoV-2 sequences and three involving out-state human SARS-CoV-2 sequences. These spillovers were detected across multiple states, involving multiple genetic lineages, including the Alpha, Gamma, Delta, and Omicron variants.

## Repeated adaptive amino acid substitutions detected in white-tailed deer

To elucidate evolutionary patterns of SARS-CoV-2 after introduction to white-tailed deer, we determined amino acid substitutions by comparing the white-tailed deer sequence(s) in the same spillover group with the potential human precursor. A total of 833 amino acid substitutions were identified at least once across the 106 spillover events described above. Of these substitutions, we defined repeated adaptive amino acid substitutions as those observed in at least two spillover events and not occurring in all previously collected human viruses from the same state in public databases (Methods). A total of 112 repeated amino acid substitutions were observed across multiple viral proteins, but predominantly in the ORF1a, ORF1b, and S proteins. Positive selection was observed in 58 of these repeated substitutions, while negative selection was observed in only one of them (Fig. 3a and Supplementary Data 3). These 58 positive selection substitutions were detected across multiple lineages in the same VOC and

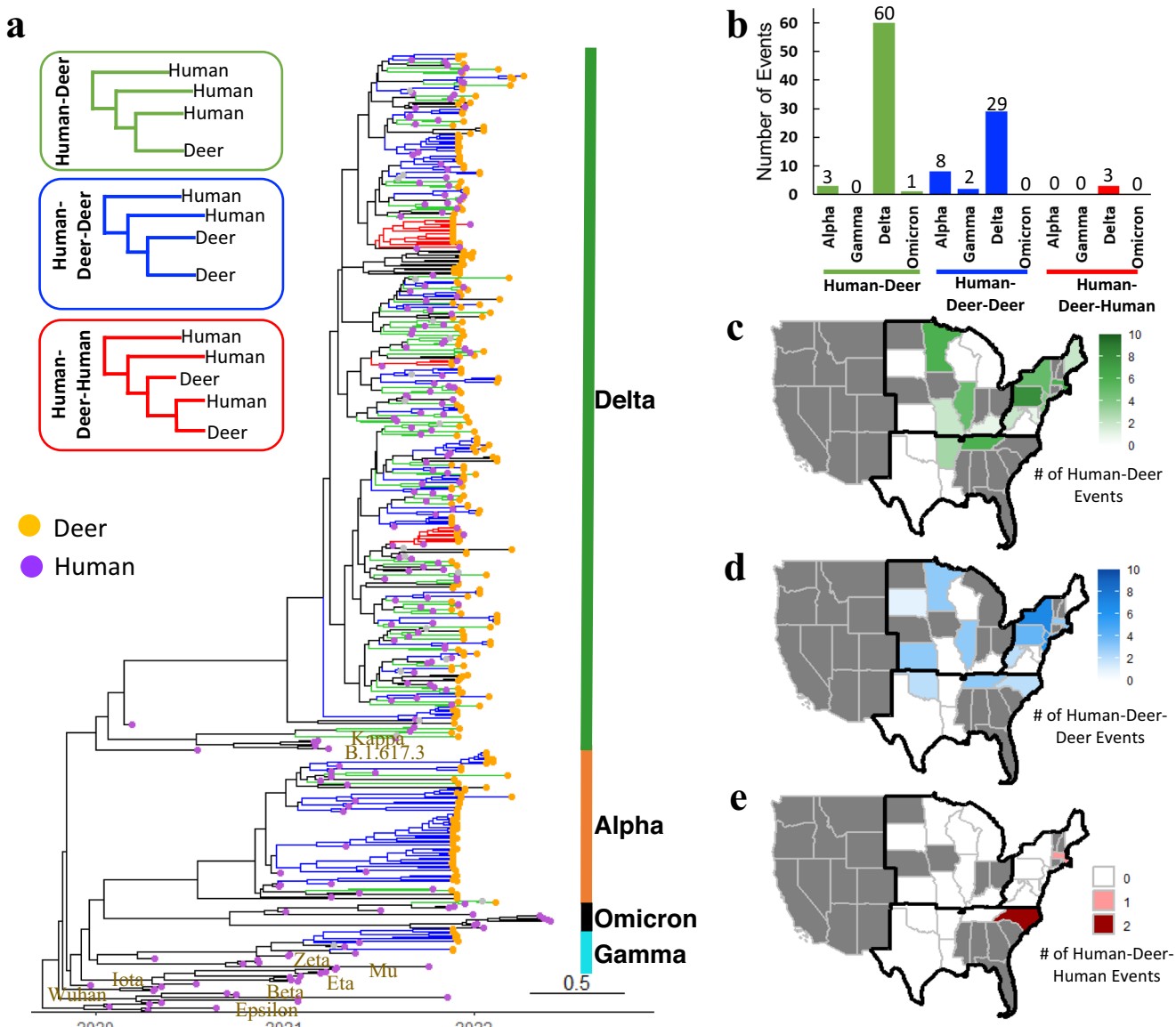

**Fig. 2 | Genetic association between SARS-CoV-2 viruses from humans and those from free-ranging white-tailed deer in the United States (November 2021-April 2022). a** The maximum clade credibility tree for white-tailed deer SARS-CoV-2 sequences ($n = 282$) and their potential precursor viruses in humans inferred 109 independent spillovers events of SARS-CoV-2 from humans (directly or indirectly) to white-tailed deer (Supplementary Data 2 and Supplementary Data 11). Three types of spillover events were identified: Human-Deer (green), where each event consists of at least one human precursor sequence and one white-tailed deer SARS-CoV-2 sequence; Human-Deer-Deer (blue), where each event consists of at least one

human precursor sequence and at least two white-tailed deer SARS-CoV-2 sequences; Human-Deer-Human (red) where each event consists of at least one human precursor sequence, at least two white-tailed deer sequences, and an additional human SARS-CoV-2 sequence. The timescale of the phylogenetic tree was represented in units of years, and the scale bar indicates the divergence time in years. **b** The number of spillover events by variant of concern. **c** Geographic distribution of the Human-Deer, **d** Human-Deer-Deer, and **e** Human-Deer-Human events. The source data for subpanel **b**–**e** is available in the Source Data file.

---

across different VOCs, and three of them were observed in more than 10 spillover events.

Of these repeated substitutions, 27 were observed in the non-structural protein (NSP) 3, which is the largest SARS-CoV-2 protein and is involved in viral replication as part of the NSP3-4-6 complex; this complex also functions in endoplasmic reticulum modification and formation of double-membrane vesicles Papain-like protease domain[21,22]. These substitutions were primarily distributed in the Ubl1, acidic domain, and Mac1, which are related to single-stranded RNA binding and interaction with the nucleocapsid (N) protein, and binding of ADP-ribose or poly(ADP-ribose)[21] (Fig. 4a). Multiple repeated substitutions were observed in NSP7, NSP8, and NSP12, which form a RNA-dependent RNA polymerase (RdRp) complex that catalyzes the

synthesis of viral RNA[23]; most of these substitutions were located outside of the RdRp structure, including one in the nucleotidyl-transferase (NiRAN) domain[24], but none were observed in or close to enzymatic sites (e.g. fingers, palm, and thumb of NSP12)[25] (Fig. 4b). In addition, 18 repeated substitutions were observed in the Spike protein with eight in N-terminal domain, one in the subdomains 1 and 2, and one in the heptapeptide repeat sequence (HR2) domain (Fig. 4c). Surprisingly, no repeated substitutions were observed in the RBD domain of Spike.

In summary, SARS-CoV-2 rapidly and repeatedly adapted to white-tailed deer with reoccurring and positively selected amino acid substitutions across the Spike protein (although not in the RBD), replicase, and other proteins.

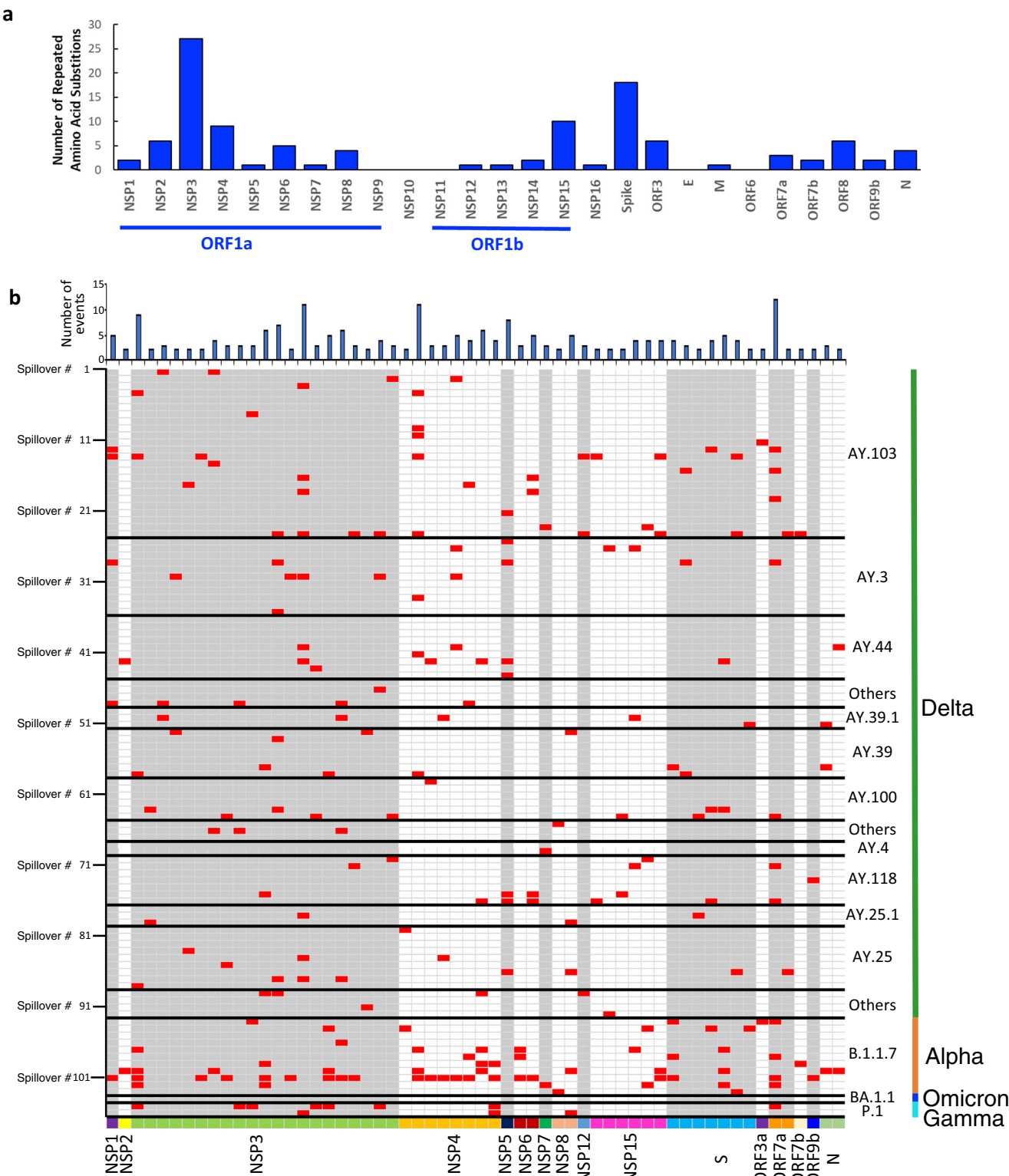

**Fig. 3 | Repeated amino acid substitutions in the SARS-CoV-2 variants detected in wild white-tailed deer populations. a** # of repeated amino acid substitutions across SARS-CoV-2 proteins; **b** repeated amino acids substitutions under positive selection and their association with the independent transmission events shown in Fig. 2. Each row represents a transmission event, and a red box in each column represents an adaptive substitution observed in a specific event (row). Supplementary Data 2 and Supplementary Data 11 contain a list of the spillover events. For additional information on amino acid substitutions, please refer to Supplementary Data 3. The source data for each subpanel is available in the Source Data file.

## Transmission of SARS-CoV-2 between white-tailed deer

To further evaluate whether SARS-CoV-2 is spreading among white-tailed deer and has become enzootic in white-tailed deer, we also assessed seroprevalence in the state of New York (NY), a state with one of the largest numbers of nasal/oral swabs and sera samples available, as well as consistent and persistent SARS-CoV-2 positivity in qRT-PCR screening. In total, we sampled nasal/oral swab and/or serum samples from 987 individual animals across 38 counties. Of them, 790 animals

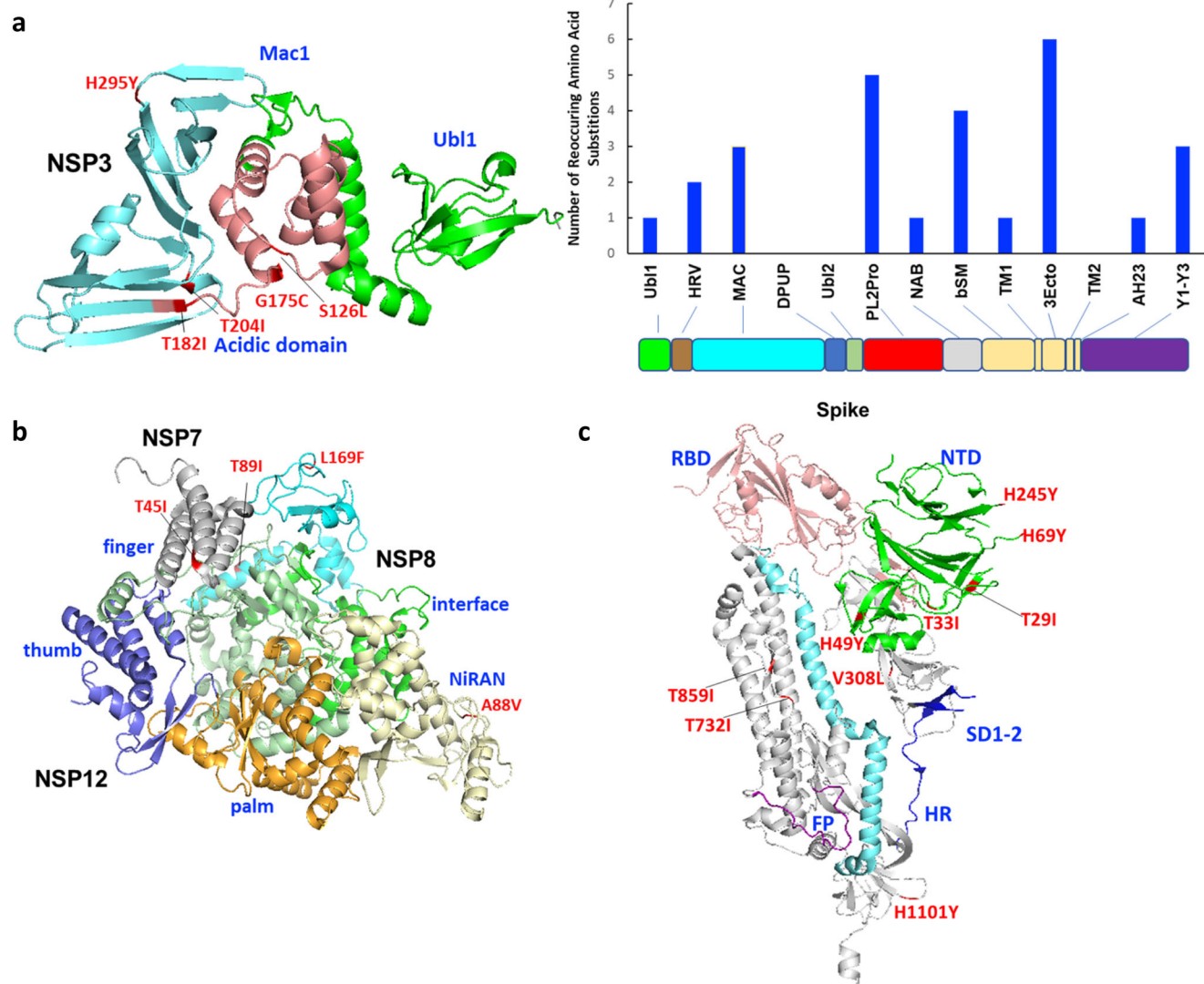

**Fig. 4 | Visualization of repeated amino acid substitutions in the SARS-CoV-2 variants detected in wild white tailed deer populations. a** Papain-like proteinase (non-structural protein 3 [NSP3]; template with Protein Data Bank [PDB] accession #6wuu), **b** RNA-dependent RNA polymerase (NSP7, NSP8, and NSP12; template with PDB #6m71), and **c** spike protein (template with PDB #6vxx). The structures were visualized by PyMOL. Additional substitutions are listed in Supplementary Data 3. The source data for the bar figure in subpanel a is available in the Source Data file.

had paired swab and sera samples, 68 had swabs only, and 129 had serum samples only (Supplementary Data 4).

Results from qRT-PCR screening showed that 184 animals (21.45%) were positive for SARS-CoV-2, with positive samples from half of the 38 sampled counties (Fig. 5a). For the six counties with ≥50 animals, county-level qRT-PCR positivity ranged from 8.70 to 62.50%. In comparison, results from surrogate virus neutralization test (sVNT) showed that 332 animals (36.12%) were seropositive for SARS-CoV-2, covering 22 sampled counties (Fig. 5a). For five counties with ≥50 animals, county-level seropositivity rate ranged from 5.26% to 55.77%. The higher antigenic prevalence for virus in nasal swabs compared to seroprevalence suggested active outbreaks in white-tailed deer during the sampling period.

Among 790 animals with paired nasal/oral swabs and serum samples from NY, 101 were SARS-CoV-2-positive by both qRT-PCR and sVNT assays, 60 only by qRT-PCR, and 168 only by sVNT. An animal was considered to have been exposed to SARS-CoV-2 when the test was positive by either qRT-PCR or sVNT, resulting in 415 out of 987 (42.05%) unique animals sampled from NY that were exposed to SARS-CoV-2.

To understand the transmission patterns between free-ranging white-tailed deer, we performed Bayesian Stochastic Search Variable Selection (BSSVS) analyses of the NY viruses from six genetic clusters, each of which involved at least four white-tailed deer infections (Fig. 5c). Results showed that viruses in four clusters spread across neighboring counties and those in two clusters were limited to the same county (Supplementary Data 5).

Taken together, our results showed that SARS-CoV-2 viruses were enzootic in free-ranging white-tailed deer populations with active transmission among populations at local levels.

## Potential spillover of white-tailed deer-adapted SARS-CoV-2 to human

Our phylogenetic analysis identified three clusters with potential secondary zoonotic transmission events, two in NC and one in MA (Fig. 2a, Human-Deer-Human clusters). We performed additional analyses, which showed that the nucleotide sequence of one SARS-CoV-2 from a human case in NC was 99.93% identical to the viruses from white-tailed deer in NC (Figs. 6a and 7), another SARS-CoV-2 from a human case in NC was 99.94% identical to the viruses from white-tailed deer in NC

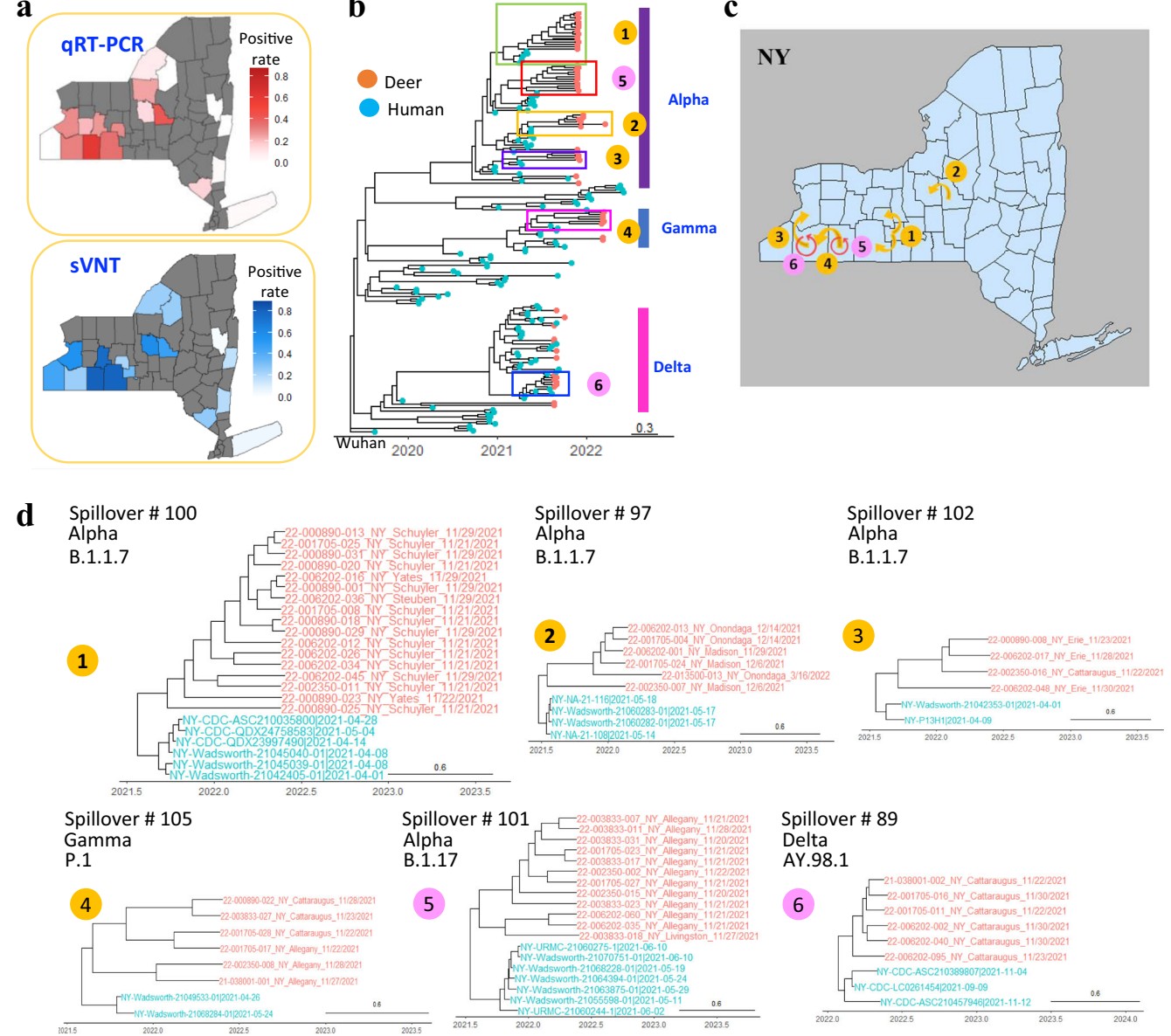

**Fig. 5 | Transmission of SARS-CoV-2 in free-ranging white-tailed deer in the state of New York (NY).** **a** County specific positive rate by quantitative reverse transcription polymerase chain reaction (qRT-PCR) or surrogate virus neutralization test (sVNT) in NY, and only those counties with at least four samples were included; **b** the maximum clade credibility tree of white-tailed deer SARS-CoV-2 viruses in NY with six transmission events with at least four deer sequences detected; **c** transmission events of SARS-CoV-2 viruses in NY analyzed by Bayesian Stochastic Search Variable Selection method; **d** the maximum clade credibility trees of white-tailed deer SARS-CoV-2 viruses associated with cross-county transmission events (case #1-4, in orange) and within-county (case # 5-6 in pink) detected by using phylogeographic analyses. The Bayes factors for these across-county transmission events are listed in Supplementary Data 5. The nodes in red were SARS-CoV-2 sequences from white-tailed deer, and those in cyan from human. The source data for panel a is available in Supplementary Data 4 and the Source Data file.

(Fig. 8). The two SARS-CoV-2 sequences from two human cases in MA were 99.96% identical to those from white-tailed deer in MA (Fig. 9).

To evaluate whether these were potential spillback events, we identified white-tailed deer-adapted sequences using the aforementioned repeated amino acid substitutions (Fig. 3), and further evaluated whether white-tailed deer-acquired substitutions may have spread to humans. Fourteen repeated amino acid substitutions were observed in white-tailed deer sequences from a NC AY.103 Human-Deer-Human phylogenetic cluster (Event #24 in Supplementary Data 2). Among them, two white-tailed deer positively adaptive substitutions, S:S680F and ORF1a:T2283I (in protein NSP3), were observed in a single human SARS-CoV-2 sequence (GSAID accession number: EPI_ISL_9246286, virus name: NC-CORVASEQ-1086-651) (Fig. 6a).

Frequencies of each of the two substitutions were negligible in other human SARS-CoV-2 sequences but high in white-tailed deer sequences (Fig. 6b). Three other white-tailed deer-specific substitutions (i.e., ORF1a:S443P [NSP2], ORF1b:V1271L [NSP 13], and ORF1a:T1678A [NSP 3]) were observed in the human NC-CORVASEQ-1086-651 sequence but infrequently in AY.103 human sequences (Fig. 7 and Supplementary Data 6). No other human SARS-CoV-2 viruses from public databases contain the combination of these five white-tailed deer-specific substitutions (Supplementary Data 6).

The human and white-tailed deer sequences in the NC AY.44 Human-Deer-Human event (Event #40 in Supplementary Data 2) had a combination of three white-tailed deer-specific repeated substitutions (S:D936H, ORF8:K68E, and N:T379I), which were not detected in any

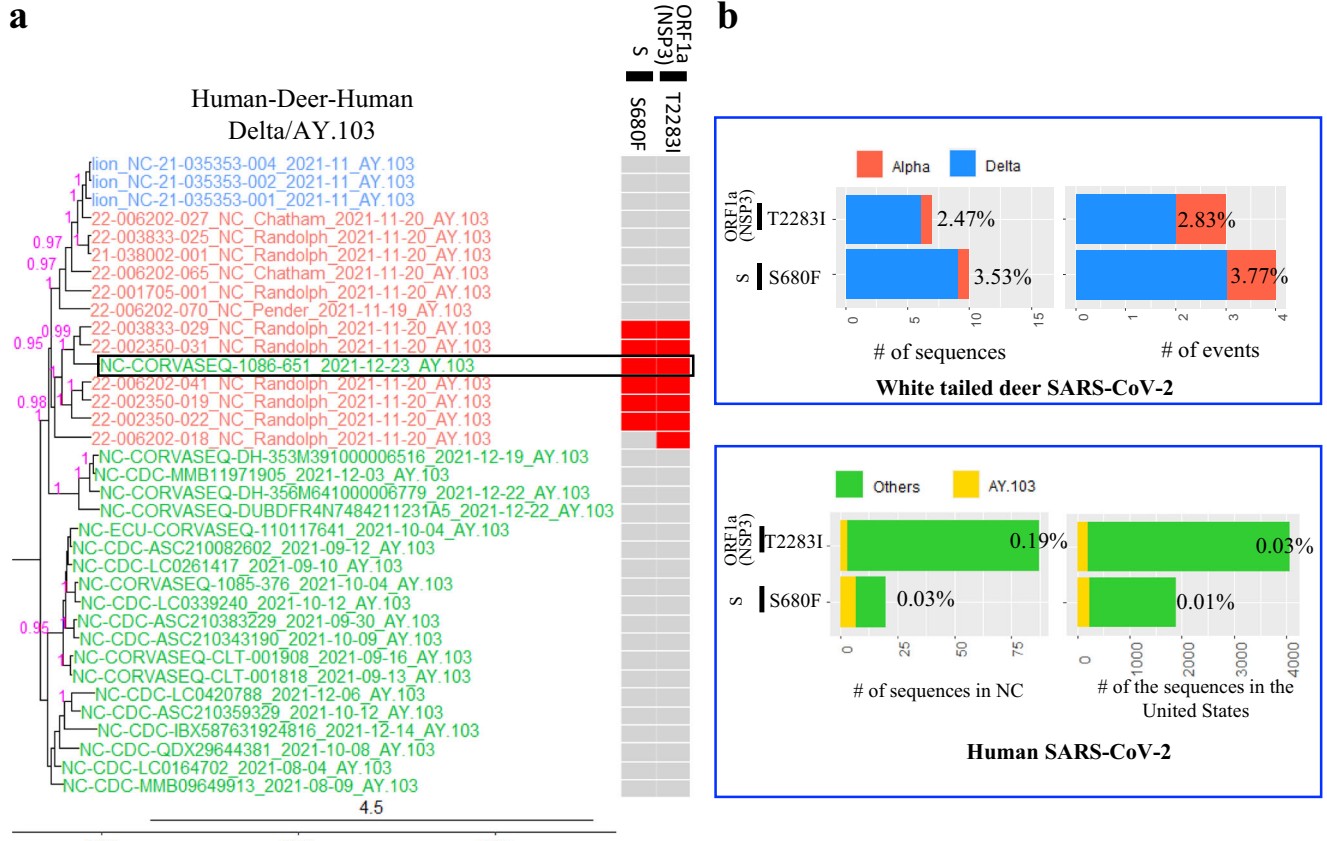

**Fig. 6 | A potential transmission event of white-tailed deer-adapted SARS-CoV-2 from white-tailed deer to humans. a** The maximum clade credibility tree illustrating the genetic relation among the white-tailed deer SARS-CoV-2 sequences and those in humans, and the human virus potentially originated from white-tailed deer is boxed; the presence of white-tailed deer-specific repeatedly adaptive amino acid substitutions were labeled in red for each node in the tree. The nodes in red were SARS-CoV-2 sequences from white-tailed deer, those in green from humans, and those in blue from zoo lions. The timescale of the phylogenetic tree was represented in units of years, and the scale bar indicates the divergence time in years. **b** frequency of the two white-tailed deer-specific repeatedly adaptive substitutions in the white-tailed deer or human SARS-CoV-2 sequences in public databases. ORF open reading frame, S spike protein, and NSP non-structural protein. The source data for subpanel b is available in the Source Data file.

other human sequences in public databases (Fig. 8). Similarly, the human and white-tailed deer sequences in the MA AY.119 Human-Deer-Human event (Event #65 in Supplementary Data 2) had a combination of three SNPs (i.e., C7303T, C21459T, and C26469T), which were not identified in any other human SARS-CoV-2 sequences in public databases (Fig. 9 and Supplementary Data 7).

To determine if any of the white-tailed deer-specific repeated amino acid substitutions could be present as intra-host single nucleotide variations (iSNVs) in human SARS-CoV-2 viruses, we analyzed the minority iSNVs, in 148 human SARS-CoV-2 viruses that we sequenced from Missouri, belonging to the same AY.103, AY.119, and AY.44 lineages as our Human-Deer-human events (Supplementary Data 8). A minority iSNV was defined when the frequency of an allele was at least 5% but less than 50% among the reads. We confirmed the absence of any SNVs at the sites where we detected white-tailed deer-specific repeated amino acid substitutions in the Human-Deer-human events.

In summary, our analyses suggested three potential spillover cases of white-tailed deer SARS-CoV-2 viruses to humans, with white-tailed deer-specific amino acid substitutions or SNPs spread to humans.

## Discussion

In this study, we demonstrated that SARS-CoV-2 has been detected as enzootic in free-ranging white-tailed deer across nearly half of the states in the US. Sequenced viruses from Alpha, Gamma, Delta, and Omicron lineages were identified (Fig. 1c), although only Delta and Omicron predominantly circulated in the human population during this sampling period. The SARS-CoV-2 lineages were still present in white-tailed deer, months after the decline of those lineages in the human population (Fig. 1d). These results were consistent with an earlier report, which showed that two white-tailed deer were infected with the Alpha variant in the middle of November, six to seven months after the Alpha wave in humans[18]. In another study, Caserta et al. detected Alpha and Gamma viruses in New York white-tailed deer between October and December 2021 after these VOCs were replaced by Delta and Omicron in humans, and detected Delta viruses in New York white-tailed deer between October and November 2021 during the period when Delta and Omicron were predominant in humans[19]. In addition, the lineages identified in this study also included those that emerged in the US during late 2021 or 2022, overlapping our sampling period. Overall, this study demonstrated that frequent introductions of new human viruses into free-ranging white-tailed deer continued to occur, and that SARS-CoV-2 VOCs were capable of persisting in white-tailed deer even after those variants became rare in the human population.

Our analyses suggested that the viruses from white-tailed deer had high genetic identities compared with those from humans, and there were at least 109 independent spillover events from humans to white-tailed deer (Fig. 2a). Given the high abundance of white-tailed deer in the US and their distribution throughout rural, suburban, and urban environments, direct and indirect interactions between human and white-tailed deer are frequent. For example, supplemental feeding of white-tailed deer has been a common source of interactions

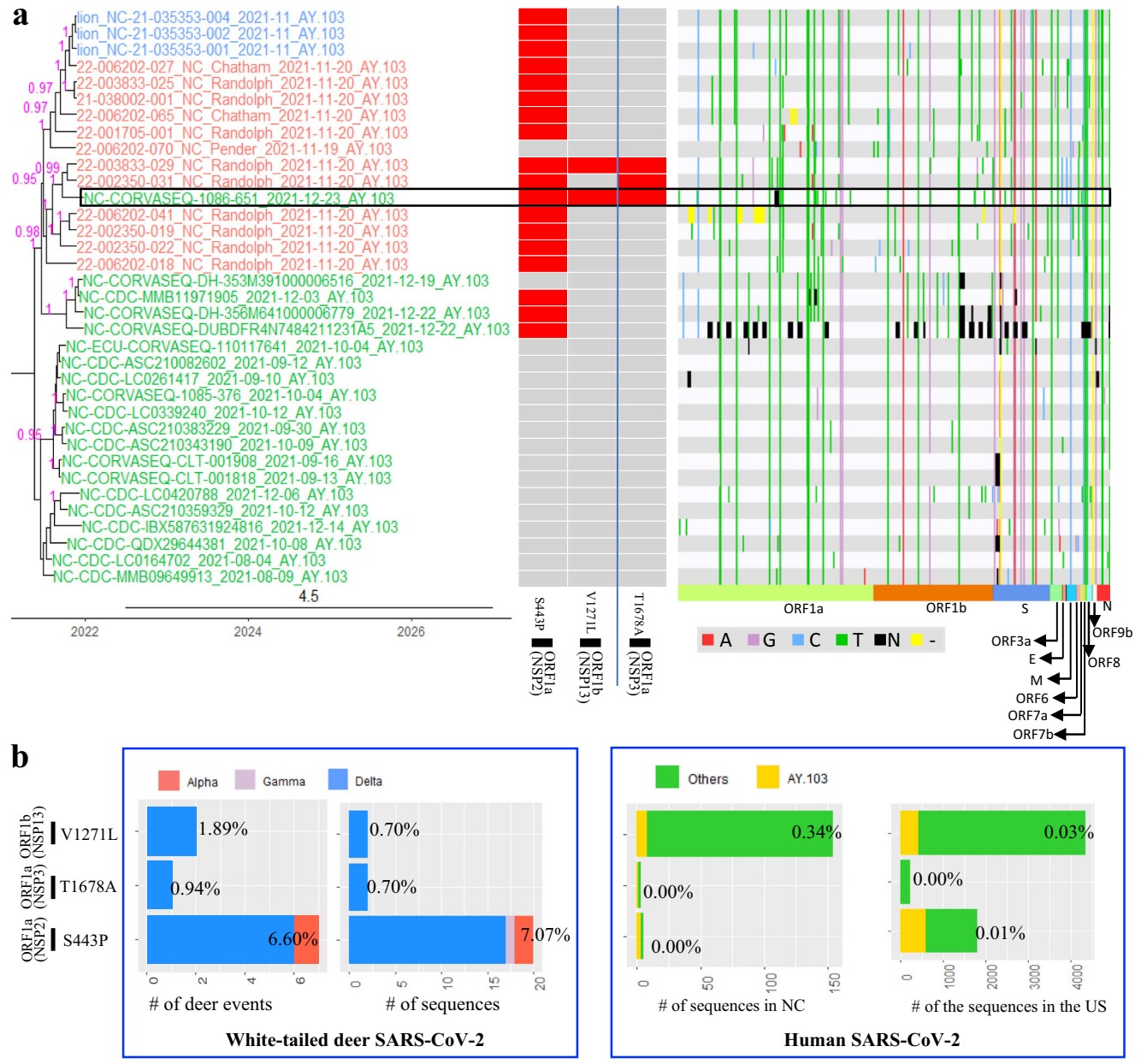

**Fig. 7 | A potential transmission event of white-tailed deer-adapted AY.103 SARS-CoV-2 from white-tailed deer to humans in North Carolina (NC). a** The maximum clade credibility tree illustrating the genetic relation among the SARS-CoV-2 sequences from white-tailed deer and those from humans, and the human SARS-CoV-2 sequence potentially originated from white-tailed deer is boxed; the genomic diversity (comparing to Wuhan-Hu-1/2019) and the presence of two white-tailed deer-specific repeated amino acid substitutions as well as one event specific amino acid substitution (divide by blue line) was labeled in red for each node in the tree. The timescale of the phylogenetic tree was represented in units of years, and the scale bar indicates the divergence time in years. **b** frequency of the three white-

tailed deer-specific substitutions in the white-tailed deer or human SARS-CoV-2 sequences. The nodes in red were SARS-CoV-2 sequences from white-tailed deer, those in green from humans, and those in blue from zoo lions. We analyzed for inter-host single nucleotide variations (iSNVs) in the 12 white-tailed deer sequences and defined a minor iSNV as having at least 5% but less than 50% prevalence among the reads. Among these 12 samples, we identified a single iSNV in 22-002350-031_NC_Randolph_2021-11-20_AY.103, which was ORF1b:V1227 1L with 461 reads coding for amino acid L (39.6%) and 703 reads coding for V (60.4%) at this position. The source data for subpanel b is available in the Source Data file.

between humans and white-tailed deer and has been documented to facilitate the transmission of bovine tuberculosis in free-ranging white-tailed deer in Michigan[26]. With highly frequent introductions of human SARS-CoV-2 into white-tailed deer, both direct and indirect transmission routes remain plausible. Further studies are needed to characterize and identify interactions that have the potential to lead to transmission between humans and white-tailed deer. Viral introduction to white-tailed deer could result from exposures to contaminated environments and fomites (e.g., contaminated human food waste, masks, and other waste products) or through direct interactions with

other animal hosts. White-tailed deer have direct and indirect interactions with multiple other animal species (e.g., red fox, skunks, and other rodents), which are susceptible to infection with SARS-CoV-2[27–30].

Since its emergence, SARS-CoV-2 has evolved rapidly in human populations. The virus has acquired a number of amino acid substitutions across the Spike and other proteins. Of these substitutions, a few significant ones have involved the Spike RBD domain, and resulted in viruses with increased receptor binding to human ACE2 receptors[31] and improved ability to evade neutralizing antibodies[32]. In addition, substitutions have also been observed in other proteins such as the

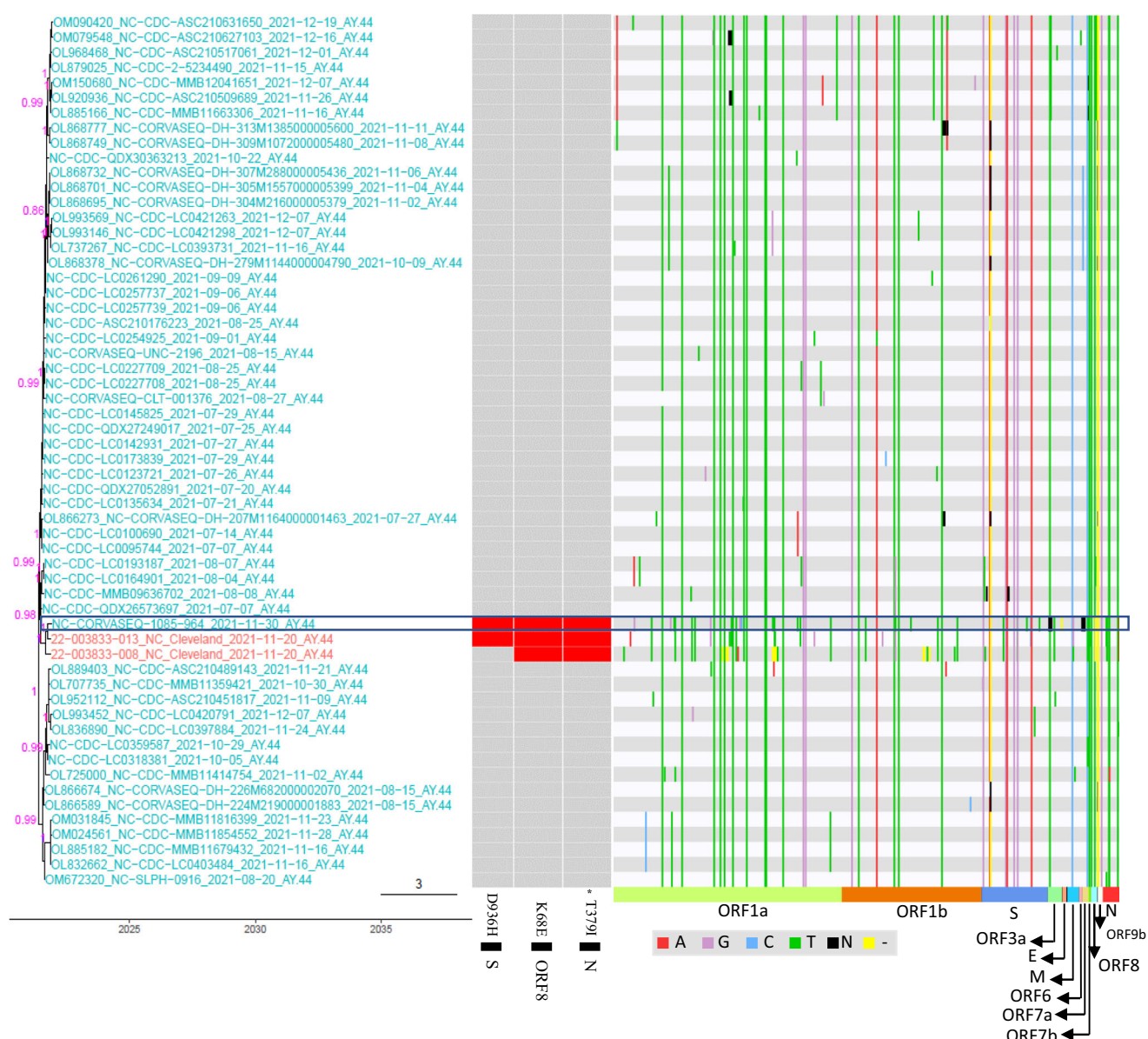

**Fig. 8 | A potential transmission event of white-tailed deer-adapted AY.44 SARS-CoV-2 from white-tailed deer to humans in North Carolina (NC).** The maximum clade credibility tree illustrating the genetic relation among the white-tailed deer SARS-CoV-2 sequences and those in humans, and the human virus potentially originated from white-tailed deer is boxed; the genomic diversity (comparing to Wuhan-Hu-1/2019) and the presence of 3 white-tailed deer-specific repeated amino acid substitutions (star marks the one under positive selection) was labeled in red for each node in the tree. No human SARS-CoV-2 viruses (except for the one listed in the box) from public databases contain the combination of these three white-tailed deer specific repeated amino acid substitutions. The nodes in red were SARS-CoV-2 sequences from white-tailed deer, and those in cyan from human. The timescale of the phylogenetic tree was represented in units of years, and the scale bar indicates the divergence time in years.

nucleoprotein (e.g., NP:R203K/G204R), which may increase the local positive charge to increase RNA binding and RNP assembly efficiency due to the additional basic amino acids[33]. In this study, we identified 112 reoccurring amino acid substitutions in white-tailed deer viruses, and 58 of those were under positive selection, suggesting that the viruses were adapting to white-tailed deer. However, no substitutions were observed in either the RBD or the polymerase enzymatic sites, indicating that the RBD and polymerase were likely already fit for white-tailed deer. The roles of these reoccurring substitutions in white-tailed deer adaptation need further investigation.

We further explored whether the 58 white-tailed deer-adaptive substitutions were situated in immunogenic regions of SARS-CoV-2 and if they overlapped with any substitutions identified in other animals or humans. As our understanding of white-tailed deer-specific

T-cell and B-cell epitopes is currently limited, we investigated whether the white-tailed deer-adaptive substitutions were located within human-specific T-cell and B-cell epitopes of SARS-CoV-2, which have been curated in the Immune Epitope Database (https://www.iedb.org). Our analysis revealed that 51 out of the 58 white-tailed deer-adaptive substitutions were situated in reported human T-cell ($n = 9$) or B-cell epitopes ($n = 42$) of SARS-CoV-2 (Supplementary Data 9). The spillover of these white-tailed deer-adapted strains back to humans has the potential to undermine the effectiveness of the pre-existing immunity associated with these epitopes generated from previous SARS-CoV-2 infections and/or vaccinations within the human population. Nevertheless, further research is necessary to comprehend the relationship between the adaptive significance of these substitutions and white-tailed deer herd immunity.

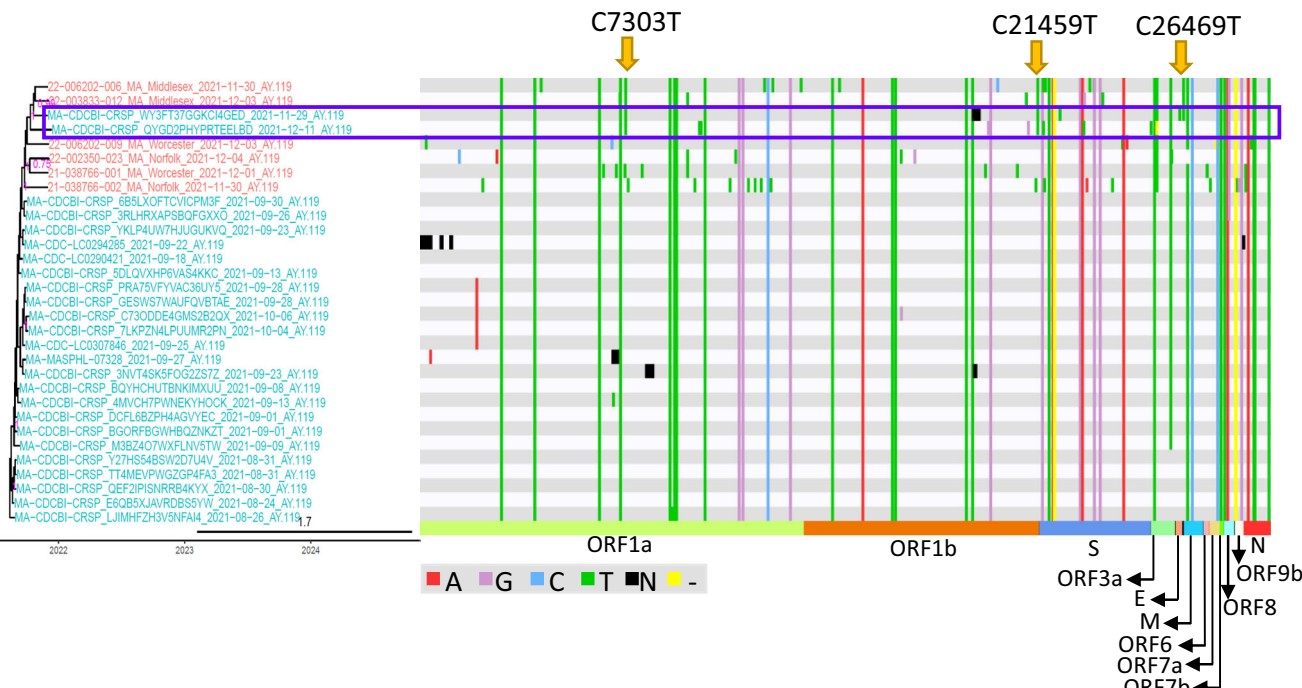

**Fig. 9 | A potential transmission event of white-tailed deer-adapted AY.119 SARS-CoV-2 from white-tailed deer to humans in Massachusetts (MA).** Three white-tailed deer specific polymorphisms was identified in the human SARS-CoV-2 viruses in box. No human SARS-CoV-2 viruses (except for the one listed in the box) from public databases contain the combination of these three polymorphisms. The frequency of each of three individual polymorphisms in other human SARS-CoV-2 viruses are listed in Supplementary Data 7. The nodes in red were SARS-CoV-2 sequences from white-tailed deer, and those in cyan from human. The timescale of the phylogenetic tree was represented in units of years, and the scale bar indicates the divergence time in years.

In contrast to white-tailed deer, a study on mink reported four substitutions (L452M, Y453F, F486L, N501T) in the spike protein during a SARS-CoV-2 outbreak on a Dutch farm[34], and three of those sites (Y453F, F486L, N501T) were observed repeatedly in mink outbreaks[35]. Bashor, et al.[36] identified 7 SNVs (H69R, D215H, D215N, N501T, D614G, H655Y, and S686G in Spike, and L37F in ORF1a) that have been reported as variants of concern in humans or other species such as cats, dogs, hamsters, and ferrets. Adaptive mutations were seen in these variants across multiple hosts. However, these sites were not among the ones we identified in white-tailed deer. In addition, we compared the positive selection sites in white-tailed deer to those reported in humans[37–39], and found that none overlapped. Thus, the adaptive substitutions that we identified in this study were likely white-tailed deer-specific and may have arisen within the deer population as a result of selective pressure on the virus.

By integrating serological and molecular data, we showed approximately 42% of white-tailed deer in NY had been exposed to SARS-CoV-2 (Fig. 5a; Supplementary Data 4). Transmission events were identified within and across counties from phylogeographic analyses (Fig. 5b, c), which strongly suggests that active transmission occurred among white-tailed deer. Of interest, 101 white-tailed deer were seropositive by both qRT-PCR and sVNT tests, as was observed during the late stages of infection in white-tailed deer experimental infection studies[13,14]. In those studies, white-tailed deer were shown to become noninfectious once neutralizing antibody titers were detectable.

Efficient deer-to-deer transmission could facilitate establishment of white-tailed deer as a reservoir of SARS-CoV-2, presenting continuous risks for zoonotic transmission back to people and to other animals. The lineages with white-tailed deer-adaptive mutations may further increase the risk to humans. In addition, transmission of older SARS-CoV-2 viruses from white-tailed deer to people that are now uncommon in the human population may pose a greater risk as immunity to these lineages wane[40,41]. Our findings highlight the potential public health implications of white-tailed deer-to-human transmission, but further studies with larger sample sizes are needed to fully understand the extent of transmission and the associated risks to human health.

Similar zoonotic transmission events have been documented in human seasonal H3N2 influenza A viruses, which spilled over from humans to pigs in the 1990s[42], facilitating the generation of the 2009 pandemic virus through genetic reassortments with contemporary human, avian, and swine influenza viruses[43]. The swine H3N2 variant (H3N2v) virus remained antigenically more similar to human H3N2 precursor viruses in the 1990s but not contemporary seasonal H3N2 viruses in humans[44]. Since 2011, this H3N2v virus has frequently been transmitted back to humans and has caused 439 confirmed infections, particularly in those who were born after 2000 and had not yet been exposed to this virus subtype[45].

Recently, a secondary zoonosis case was reported in Canada that was associated with a highly distinct lineage sequenced from white-tailed deer[11]. To test whether there were any secondary zoonosis events among our samples, we comprehensively compared our white-tailed deer SARS-CoV-2 sequences and all human SARS-CoV-2 sequences in public databases. We identified three potential cases of reverse zoonosis in two US states. A retrospective epidemiologic investigation was conducted to determine whether the five people associated with the AY.103 Human-Deer-Human event in NC, which also included three SARS-CoV-2 sequences from zoo lions (Fig. 6a), had contact with deer in the month prior to the associated illness with COVID-19. We successfully contacted three individuals, but none reported close contact with either deer or the zoo. Nevertheless, the viruses in the human cases from all three deer to human transmission events we detected had white-tailed deer-specific amino acid substitutions or SNPs, which were negligibly observed in other human SARS-CoV-2 genomes. Inferring secondary zoonosis from a phylogenetic tree is limited by available sequence data. Information is currently scarce regarding how often infected humans are in contact with white-tailed deer or vice versa, and how likely cross-species transmission occurs under those circumstances. Overall, these potential

instances highlight the possibility for bi-directional transmission of SARS-CoV-2 at the human-white-tailed deer interface.

A limitation of this study is that the human SARS-CoV-2 data from public databases lack geographic granularity and may not cover the areas where our white-tailed deer samples were collected, although over 14 million genomes sequences were included in our analyses. For example, among 282 white-tailed deer SARS-CoV-2 sequences, there were 18 which we could not identify any genetically close human SARS-CoV-2 sequences within the same state as the potential precursor virus (Fig. 2a). This study also has a limitation in that the number of samples collected from the same sampling site was limited, typically spanning less than two weeks. This limited sampling duration prevented us from fully understanding the duration and population dynamics of each SARS-CoV-2 variant in the white-tailed deer population. Additional research analyzing the white-tailed deer and human SARS-CoV-2 viruses collected over an extended period from the same geographic area is needed to further elucidate the transmission patterns of SARS-CoV-2 in white-tailed deer and between people and white-tailed deer.

In summary, this study shows that SARS-CoV-2 was enzootic in white-tailed deer, and the viruses circulating in white-tailed deer stem from frequent and independent spillover events from humans with multiple genetic lineages co-circulating among white-tailed deer, including lineages observed in humans prior to and during our sampling period. Continued large-scale surveillance of white-tailed deer is necessary to understand the evolution and distribution of genetic variants in white-tailed deer, evaluate whether the white-tailed deer are a potential reservoir for SARS-CoV-2 viruses, and the role of white-tailed deer in ecology and natural history of SARS-CoV-2.

## Methods

### Ethics statement

White-tailed deer were captured under a wildlife damage management agreement administered by USDA/APHIS Wildlife Services. Sample collection was conducted opportunistically in collaboration with regulatory agencies and as part of routine surveillance programs. Sex and gender were not excluded in this study. This study focuses on viral evolution and transmission and did not include an analysis of host gender and sex.

To compare intra-host intra-host single nucleotide variations between human and white-tailed deer SARS-CoV-2 viruses, the de-identified human samples testing positive for SARS-CoV-2 were selected from a cohort of routinely collected samples at University of Missouri Health Care which covers the Columbia, MO and the neighboring counties, and this study was approved by the University of Missouri Institutional Review Board (#2025449). Sex and gender were not excluded in this study. No targeted recruitment efforts were undertaken.

### White-tailed deer sample collection

From November 4, 2021 to April 4, 2022, a total of 8830 white-tailed deer nasal swab samples were collected from Washington, D.C. and 26 participating states in the US either by hunter- or agency-harvest. None of the animals sampled exhibited clinical signs of SARS-CoV-2 infection. Over 95% of the white-tailed deer population resides in the Northeast, Midwest, and Southeastern United States, represented by Washington, D.C. and 26 participating states in the US from which the samples were collected[46]. These regions are also where much of the 6 million white-tailed deer are harvested annually by hunters[47]. In the vast majority of cases, a paired blood sample was collected onto a Nobuto filter strip for serological analysis.

### RNA extraction and qRT-PCR

SARS-CoV-2 RNA was prepared from oral and nasal swab samples preserved in PrimeStore Molecular Transport Media (MTM, Longhorn Vaccines and Diagnostics LLC, catalog# LH105) using MagMAX™ CORE Nucleic Acid Purification Kits (Applied Biosystems, catalog# A32702) in accordance with the manufacturers' instructions. 5 μL of RNA extract

was for qRT-PCR detection of SARS-CoV-2 N1 and N2 targets using the BioRad Reliance One-Step Supermix Kit (catalog# 12010221) with SARS-CoV-2 RUO Primers & Probes obtained from Integrated DNA Technologies (catalog# 10006713). Reaction and thermocycling conditions were identical to those described for the BioRad Reliance SARS-CoV-2 RT-PCR assay, and data was acquired BioRad CFX96 Touch Real-Time PCR Detection System or CFX Opus Real-Time PCR System.

### Genomic sequencing and assembly

For SARS-CoV-2 whole genome RT-PCR amplification, cDNA libraries were prepared using the Nextera XT DNA Sample Preparation Kit (catalog# FC-131-1096), and sequencing was performed using the 500 cycle MiSeq Reagent Kit v2 (catalog# MS-102-2003) according to manufacturer instructions[48]. The quality of paired-end reads obtained from MiSeq sequencing were analyzed and trimmed with a Phred quality score of 20, which indicates a base call accuracy of 99%, the likelihood of finding 1 incorrect base call among 100 bases[49]. The sequence assembly and consensus sequence construction were conducted with the Wuhan-Hu-1 (Accession Number: NC_045512.2) as the reference genome by using the Iterative Refinement Meta-Assembler (IRMA, v1.0.2)[50]. Manual curation and validation were performed for those genomic positions with low coverage. IRMA was used to analyze single nucleotide variants (SNVs) and assess intra-host viral genomic diversity. A minority iSNV was identified when the frequency of an allele was at least 5% but less than 50% among the reads. In addition, the consensus sequences were validated by using Qiagen CLC Genomics Workbench 20.0.4 with a quality score of 0.05.

### Genomic sequence alignment and molecular characterization

NextStrain[51] was used to align SARS-CoV-2 genomic sequences and also to identify the nucleotide and amino acid substitutions between the white-tailed deer SARS-CoV-2 sequences and the associated human precursor SARS-CoV-2 sequence. Because of low coverage at the 5' untranslated region (before position 266) and 3' untranslated region (after position 29,674) of the genome, we excluded these positions from nucleotide and amino acid substitution analyses.

### PANGOLIN lineage classification

The Phylogenetic Assignment of Named Global Outbreak Lineages (PANGOLIN) software (v4.0.5)[20] was used (PANGO v4.0.6 (2022-04-22)) to determine Pango lineages for each white-tailed deer sequence. Genetic lineage classification was achieved for 355 sequences. Among these samples, only those with a high sequencing coverage over >95% of the reference genome (n = 282), an IRMA score of 95%, were selected for further evolutionary analyses.

### Bayesian phylogenetic analyses

Time-scaled Bayesian analyses were performed by using the Markov chain Monte Carlo (MCMC) method with Bayesian phylogenetic analysis by sampling trees (BEAST)[52] v.1.10.4 software. Each tree was generated by a HKY substitution model with gamma = 4, and a coalescent exponential growth prior with strict clock. The MCMC chain length was set as 30 to 100 million iterations with subsampling every 10,000 iterations. The babette[53] R package v.2.3.2 was used for batch BEAST processing. Tracer v1.7.2 was used to assess the results. The maximum clade credibility (MCC) tree was summarized by using TreeAnnotator[54] v.1.10.4, with a burn-in rate of 20% and the estimated time of divergence being represented by the median node height. The ggtree[55] R package v3.8.0 was used for tree visualization.

### Phylogeographic analyses

To determine whether SARS-CoV-2 was transmitted among white-tailed deer populations within the same geographic area or across different geographic areas, we performed Bayesian phylogeographic analyses by using BEAST[52] (v.1.10.4). The Bayesian Stochastic Search Variable

Selection (BSSVS) analysis was performed with county as discrete traits for the white-tailed deer SARS-CoV-2 sequences. Transmission events were filtered for statistical significance using the criteria of Bayes factor[56] and posterior probability >0.7, as well as geographically nearby counties. Different statistical support levels were defined as follows: 3 ≤ Bayes factor ≤ 10 indicates support; 10 ≤ Bayes factor ≤ 100 indicates strong support; 100 ≤ Bayes factor ≤ 1000 indicates very strong support; and Bayes factor ≥ 1000 indicates decisive support.

### Identify potential precursor virus for white-tailed deer SARS-CoV-2 viruses

To identify potential precursor virus for a white-tailed deer SARS-CoV-2 virus, we grouped all human and white-tailed deer SARS-CoV-2 genomes by state and Pango lineages. As a result, 89 datasets were obtained, each containing human and white-tailed deer SARS-CoV-2 sequences from the same state and the same Pango lineage. To ensure the robustness of our selection, two methods were used to select the closest sequences for each white-tailed deer SARS-CoV-2 sequence: 1) FastTree[57] v1.4.4 was used to identify those sequences based on topology; 2) Complete Composition Vector (CCV) v1.0 method[58] was used to identify those sequences based genetic distances, and CCV is an alignment free methods to enable genetic distance measurement at large scale. The unique sequences from top 20 ranked sequences from each method were identified (Supplementary Data 10) and used for Bayesian phylogenetic analyses. Biopython v1.79 package[59] were used for sequences data processing. The following criteria were used to determine whether a human SARS-CoV-2 genome was a precursor virus for a testing white-tailed deer SARS-CoV genome: 1) the genomes shall belong to the same genetic cluster with posterior probability ≥0.70; 2) the nucleotide identity shall be at least 99.85%. To ensure our analyses will not rule out SARS-CoV-2 viruses that could be from those viruses circulating in white-tailed deer (e.g., reported by previous studies) and other animals, we performed similar analyses including viruses from non-human hosts. Results showed that the white-tailed deer SARS-CoV-2 sequences from this study are genetically diverse from those reported earlier (Fig. 10). In total, we identified a human precursor virus for 264 out of the 282 white-tailed deer SARS-CoV-2 sequences analyzed, while no human precursor virus from the same state was identified for the remaining 18 sequences.

It is possible that transmission could occur between white-tailed deer and out-of-state travelers, such as through hunting or contact with animals. To ensure inclusion of potential out-of-state transmission events, we used Ultrafast Sample placement on Existing tRees (UShER) on all 14.3 million public SARS-CoV-2 genomic data as of March 30, 2023. UShER implements the Fitch-Sankoff algorithm to infer the placement of mutations on a given tree and on the variant list and mutation-annotated tree[60]. For 264 white-tailed deer sequences with a matching human sequence from the same state in our above analyses, UShER identified genetically close virus sequences for 230 from the same state and 30 from out-of-state human sequences. For the 18 white-tailed deer sequences without any matching human sequences from the same state in our previous analyses, UShER identified genetically close sequence viruses for all of them.

The phylogenetic analysis of spillover events was performed using a combination of out-of-state human SARS-CoV-2 sequences obtained from UShER and within-state human sequences obtained from FastTree and CCV analyses. The sequences were selected based on the following criteria: (1) they belonged to the same Pango lineages as the matched white-tailed deer sequences; (2) they were collected only in the US; and (3) their collection date was either earlier than or the same as the collection date of the white-tailed deer samples.

### Identify independent spillover events from humans and transmission events within the white-tailed deer population

To identify independent potential spillover events of SARS-CoV-2 from humans and understand the transmission dynamics of SARS-CoV-2

viruses in the white-tailed deer populations, three types of clusters were defined, Human-Deer, Human-Deer-Deer, and Human-Deer-Human. For all these clusters, at least one human precursor virus was identified; those without human precursor virus were excluded from this analysis. For 282 white-tailed deer SARS-CoV-2 sequences identified, we were able to assign 238 of them to one of these three clusters (Supplementary Data 2). The following specific criteria are used for each cluster.

Human-Deer: 1) the human precursor virus and a single white-tailed deer SARS-CoV-2 sequence; 2) all human and SARS-CoV-2 sequences were from the same state and belonged to the same Pango lineage; 3) the posterior probability (PP) > 0.7 for the human-deer branch; and 4) the nucleotide sequence identities between the human precursor sequence and the white-tailed deer SARS-CoV sequence was ≥99.85%.

Human-Deer-Deer: 1) the human precursor virus and at least two white-tailed deer SARS-CoV-2 sequences; 2) all human and SARS-CoV-2 sequences were from the same state and belonged to the same Pango lineage (single white-tailed deer SARS-CoV-2 sequence from a different state inside would be removed from this event); 3) the posterior probability (PP) > 0.7 for the human-deer branch; and 4) the nucleotide sequence identities between the human precursor SARS-CoV sequence and at least one of the white-tailed deer SARS-CoV sequences ≥99.85%.

Human-Deer-Human: 1) the human precursor virus, at least two white-tailed deer SARS-CoV sequences, and another human SARS-CoV-2 sequence; 2) all human and SARS-CoV-2 sequences were from the same state and belonged to the same Pango lineage; 3) the human2 and at least one of the white-tailed deer SARS-CoV-2 sequences formed a deer-human2 sub-branch; 4) PP > 0.7 for both human1-the deer branch and the deer-human2 subbranch; 4) the nucleotide sequence identity between human1 and at least one of white-tailed deer SARS-CoV-2, and that between human2 and at least one of white-tailed deer SARS-CoV-2 was ≥99.85%.

### Positive and negative selection analyses

Positive selection occurs when the rate of nonsynonymous substitutions ($\beta$) is greater than the rate of synonymous substitutions ($\alpha$), while negative selection occurs when $\beta < \alpha$. Both positive and negative selection analyses were performed using the FUBAR (Fast, Unconstrained Bayesian AppRoximation), from the HyPhy software v2.5.42(MP)[61,62]. The phylogenetic trees generated by FastTree of each gene, were used in the analyses with the white-tailed deer sequences marked to test positive or negative selection in a specific branch in the phylogenetic tree. We considered sites with a posterior probability >0.9 to be significant, indicating either positive selection (prob($\alpha < \beta$)) or negative selection (prob($\alpha > \beta$))[61].

### Identification of repeated amino acid substitutions in white-tailed deer sequences

To understand whether amino acid substitutions could occur independently after the virus was introduced to the white-tailed deer population, we attempted to identify those repeated amino acid substitutions across the clusters of Human-Deer, Human-Deer-Deer, and Human-Deer-Human. A repeated amino acid substitution was defined with the following criteria: 1) the substitution was observed at least twice across those 106 within-state clusters of Human-Deer, Human-Deer-Deer, and Human-Deer-Human; 2) the substitution was not shown in all human SARS-CoV-2 viruses in the same lineage from the same state; 3) the substitution had a low frequency (e.g., <0.15%, obtained by Outbreak.info R package v0.2.0[63]) in all SARS-CoV-2 human viruses from the US and worldwide. If a repeated amino acid substitution was under positive selection, it was defined as a white-tailed deer-adaptive substitution.

### Parameter optimization for evolutionary analyses

To improve the accuracy and reliability of our analyses and mitigate potential issues arising from specific genomic positions that are prone

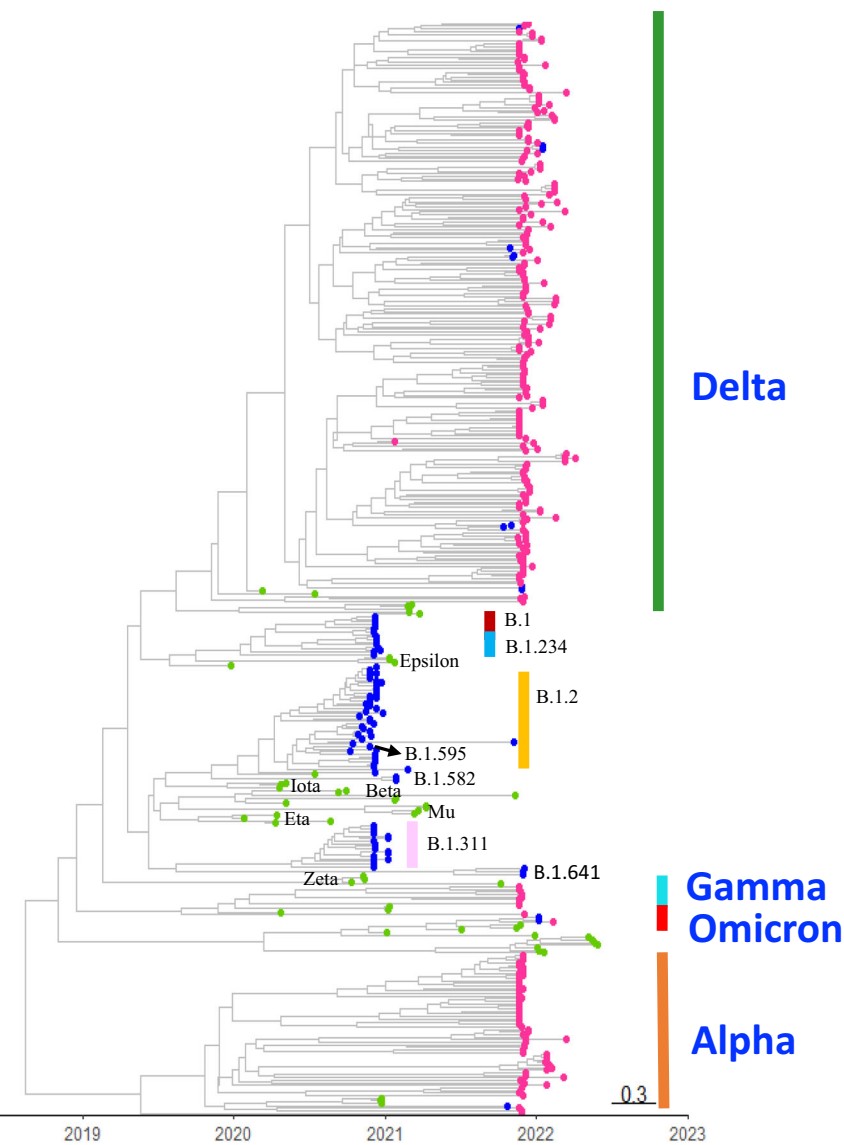

**Fig. 10 | The maximum clade credibility tree illustrating the evolutionary relation between the white-tailed deer SARS-CoV-2 sequences collected from this study and those from public database.** A total of 332 white-tailed deer SARS-CoV-2 genomes were downloaded from GISAID (Global Initiative on Sharing All Influenza Data) on December 5, 2022. Among these genomes, 118 had complete and high coverage sequences (>99% coverage) and were included in the evolutionary analyses along with the white-tailed deer SARS-CoV-2 samples collected in this study. The timescale of the phylogenetic tree was represented in units of years, and the scale bar indicates the divergence time in years.

to sequencing errors and recombination, we explored a masking approach to refine our analyses. Specifically, we masked 265 problematic positions as reported (source: https://github.com/W-L/ProblematicSites_SARS-CoV2), followed by phylogenetic analyses. The resulting tree topologies derived from the complete genome tree (see above description) and the tree with the masked positions exhibited a striking similarity (Supplementary Data 12). By excluding the reported positions from the tree, we performed a reanalysis of the transmission events, and the number of transmission events remained unchanged when using the same identification rules. We also conducted an assessment and determined that none of the white-tailed deer-specific adaptive substitutions were found within the 265 problematic positions. Therefore, it appears that the masked positions do not significantly impact the analyses conducted in this study. Therefore, all analyses conducted in this study were carried out without applying a mask to the 265 positions mentioned.

The genomic sequences of the majority of the white-tailed deer SARS-CoV-2 viruses and their corresponding potential human SARS-CoV-2 precursor viruses obtained from public databases demonstrated a high genomic nucleotide sequence identity, surpassing 99.80% (Supplementary Fig. 2a). To assess the reliability of the parameters utilized in defining transmission events, namely sequence identity and posterior probability, we conducted additional analyses encompassing various parameter ranges. Illustrated in Supplementary Fig. 2b–e, our findings revealed that the number of spillover events remained quite consistent when employing a sequence identity cutoff

of 99.85% or lower. For instance, we detected a total of 110 spillover events with a sequence identity of 99.80% or lower. However, as the sequence identity cutoff became more stringent, the number of clusters decreased notably due to the lack of corresponding human SARS-CoV-2 precursor viruses. Specifically, we identified 100 spillover events with a sequence identity of 99.90% and only 66 events with a sequence identity of 99.95%. Conversely, the posterior probability did not have a notable impact on the number of transmission events, as the tree subclades associated with the majority of these events exhibited a posterior probability of 0.90 or higher. Only the subclade associated with one human-Deer-human spillover event had a posterior probability of 0.78 and another one with human-Deer spillover event had a posterior probability of 0.66. These parameter tuning analyses further confirm the robustness of the criteria employed in the analysis of transmission events. Specifically, a sequence identity cutoff of 99.85% and a posterior probability of 0.70 have been validated as reliable and effective thresholds.

### Serological assays

Antibodies were extracted from Nobuto filter paper strips (Sterlitech, catalog # 49010010) and screened at the United States Department of Agriculture, National Wildlife Research Center (Fort Collins, USA) at a functional dilution of 1:20. Extracted samples were screened using a surrogate virus neutralization test (sVNT, Genscript cPass™, catalog# L00847-A) with data acquired using a VarioSkan Flash or Varioskan LUX multimode microplate reader (Thermo Fisher)[16]. At least two technical replicates were used for the calculation of the average % inhibition. The sVNT has not been validated for deer; however, previous evaluations with white-tailed deer sera samples suggested that sVNT results were qualitatively similar to a highly specific SARS-CoV-2 virus neutralization test[16].

### Structural modeling

Structural templates of different functions with genes of the SARS-COV-2 were downloaded from the Protein Data Bank (PDB)[64]. White-tailed deer sequences were aligned with the template sequences by MUSCLE v5.1[65]. Visualization was conducted by PyMOL v2.5.4[66].

### Public datasets

All available SARS-CoV-2 genomic sequences ($n = 11,778,398$ by 2022/07/09) from humans were downloaded from GISAID, and additional genomes ($n = 1,020,486$ by 2022/04/07) were curated from GenBank. From these sequences, human SARS-CoV sequences were selected from the 23 states where we sampled the white-tailed deer sequences. After removing redundant entries and selecting the complete and high coverage sequences, a total of 717,717 SARS-CoV-2 genomic sequences were obtained for this study. In addition, on December 5, 2022, we downloaded a total of 332 white-tailed deer SARS-CoV-2 genomes from GISAID. Among these genomes, 118 had complete and high coverage sequences (>99% coverage) and were included in the evolutionary analyses along with the white-tailed deer SARS-CoV-2 samples collected in this study. All these publicly available sequences and associated metadata used in this dataset are published in GISAID's EpiCoV database and NCBI SARS-CoV-2 Resources.

### Reporting summary

Further information on research design is available in the Nature Portfolio Reporting Summary linked to this article.

## Data availability

The genomic data of all SARS-CoV-2 viruses generated in this study have been submitted to GISAID and NCBI GenBank. Supplementary Data 1 contains the GISAID and Bioproject accession numbers and related metadata for these sequences. In the case of human SARS-CoV-2 sequences, their associated strain names and GISAID accession numbers can be found in Supplementary Data 8. To access the GISAID database, users need to log in following the instructions provided by the GISAID database. Once logged in, the GISAID database enables users to search and retrieve sequence and metadata data using either a specific accession number or a specific strain name. Additionally, other public data utilized in this study can be accessed from GISAID and GenBank, NCBI. Additionally, the original data utilized for generating bar graphs and geospatial visualizations can be accessed in the Source Data file. Source data are provided with this paper.

## Code availability

The source codes for the scripts used for molecular analyses are available at https://github.com/FluSysBio/WTD_SARS-CoV-2_Transmission and can also be accessed at https://doi.org/10.5281/zenodo.8010758[67].

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

## Acknowledgements

This project is funded by the American Rescue Plan Act provision to conduct monitoring and surveillance of susceptible animals for SARS-CoV-2 (Grant number: AP22WSNWRC00C019 to X.F.W.). C.Y.T. was supported by a training grant by the National Institute of Allergy and Infectious Diseases of the National Institutes of Health under award number F30AI172230. We thank the federal employees at USDA APHIS Wildlife Services, and collaborators at state wildlife agencies for contributing wildlife sampling expertise, as well as hunters for participating in this large-scale effort. We would also like to acknowledge Robert Pleszewski, Christopher Quintanal, Joshua Eckery, and Jason Klemm from the USDA APHIS National Wildlife Research Center, for laboratory screening of swab and Nobuto samples. We are grateful for Rich Chipman, Dennis Kohler, Derek Collins, Kelsey Weir, Tim Linder, Jourdan Ringenberg and Jon Heale for their assistance in project development and implementation and for Kim Pepin, Joshua Hewitt, and Cheng Gao for their critical discussion. In addition, we would like to thank Dr. Jun Hang and Tao Li for sequencing human SARS-CoV-2 viruses used in this study, and Kritika Prasai for generating the bam files used for polymorphism analyses. The findings and conclusions in this report are those of the authors and do not necessarily represent the official position of the U.S. Government.

## Author contributions

Conceptualization: T.J.D., S.S., J.C., M.T., and X.F.W. Methodology: A.F., J.C., A.R., K.L., M.T., X.F.W. Investigation: A.F., S.B., J.C., T.J.D., R.G., K.L., J.L., A.R., S.S., C.Y.T., S.S.T., M.T., A.U., X.F.W. Visualization: A.F., T.J.D., X.F.W. Funding acquisition: T.J.D., J.L., X.F.W. Project administration: T.J.D. and X.F.W. Supervision: T.J.D., J.L., S.B., J.C., S.S.T., M.T., X.F.W. Writing, original draft: A.F. and X.F.W. Writing, review and editing: A.F., S.B., J.C., T.J.D., R.G., A.R., S.S., C.Y.T., S.S.T., X.F.W.

## Competing interests

The authors declare no competing interests.
