## [Peer Review File · Nature Communications]

Transmission of SARS-CoV-2 in free-ranging white-tailed deer in the United StatesReviewers' Comments:

Reviewer #1:

Remarks to the Author:

Overall the authors collected a significant amount of samples of SARS-CoV-2 in white tailed deer across a large geographic range. Using this data their phylogenetic analysis find convincing evidence of many spillovers to WTD and some spillovers back to humans. This study provides important information to the state of SARS-CoV-2 in wild deer populations and a potential for public health ramifications from WTD as a SARS-CoV-2 reservoir. I do have a few suggestions for improvement and requests for clarification which I have listed below:

Major comments

- 1) Identification of spillovers was done using sequences (human and deer) from only within a state. It is unclear that it is justified to partition data in this way. In particular this criteria could exclude the genetically closest human virus thus undermining the within WTD analyses, and limit the potential spillovers detected.
- 2) Of the sites with positive selection associated with transmission in WTD, do these overlap with sites that are under positive selection associated with transmission in humans, and other animals (ex. mink)? Interpretation of this overlap (or lack of overlap) could be informative.
- 3) Fig. E3 seems to indicate some reversions of the repeated amino acid mutations of interest, it would be beneficial to discuss this behavior, perhaps clarifying the patterns with mutations mapped onto a maximum likelihood tree.

Minor comments

- 1) In the results, line 155-159 appears to be referring to sVN which is introduced at line 161. This should be clarified.
- 2) Line 156 states the number of animals that were seropositive, what percentage of tests is this?
- 3) The last 2 paragraphs of the results are somewhat difficult to follow. It was hard for me as a reader to keep track of the different clusters, perhaps name them clusters 1, 2, 3. It was also unclear based on the text, if the identified human samples were genetically linked or not (beyond the listed amino acids).
- 4) Line 204, the phrasing of "SARS-CoV-2 is enzootic in free-ranging WTD across nearly half of the states" seems imply that it is ruled out as being enzootic in other states, which is not the case, so I would suggest rephrasing.
- 5) Line 218-219, (this is likely beyond the scope of the study but would be very important to consider) is it possible to report on the symptoms of the deer, in particular if there were differences in positive and negative individuals.
- 6) Line 230 appears to be missing a word
- 7) Line 404: were sites only labeled as WTD adaptive if they were repeated and under positive selection? If this is indeed the case were sites under positive selection alone considered?
- 8) In figure 1d, when was the B.1 sample collected?
- 9) All trees should have a scale indicator

Reviewer #2:

Remarks to the Author:

The manuscript reports interesting and important results supporting previous suggestions of WTD as reservoirs for SARS-CoV-2 variants of concern, with a large number of samples and providing evidences of spillover from human to deer and spillback from deer to human. The number of tested samples collected throughout the U.S, the number of genomes retrieved and the analyses providing further evidence of spillback to humans are differentials to the established literature.

Lines 44-45: The article "From Deer-to-Deer: SARS-CoV-2 is efficiently transmitted and presents broad tissue tropism and replication sites in white-tailed deer" (Martins et al., 2022) should also be cited here. It is one of the most important on the infection and transmission dynamics of SARS-CoV-2

in WTD.

It is not clear in the methods or results if the samples used in this study were selected among a larger number of samples and how or if these states were selected for sampling.

Extended Data Figure E6: The phylogenetic tree does not contain all the WTD sequences from public databases. Otherwise, it should be mentioned how these 118 WTD sequences were selected to build this tree.

Reviewer #3:

Remarks to the Author:

Numerous groups have demonstrated that white-tailed deer (WTD) are susceptible to SARS-CoV-2 and may be a source for zoonotic infection of humans. The data provided in this work supports these prior studies and goes on to suggest that SARS-CoV-2 lineages can not only be introduced but can become enzootic and co-circulate in WTD. Specifically, the authors use a combination of sequence analysis and sero-epidemiology to demonstrate transmission within WTD populations. This data is compelling and will be important to monitor to better understand viral evolution within WTD populations within the US. My primary concerns relate to the data suggesting reverse zoonosis infection of WTD-adapted viruses to humans. Specific questions:

1. Without corresponding human sequences from the same geographic location of the deer, how can the authors "prove" that these changes weren't common in humans and spilling over into WTD?
2. The WTD-human conclusions are based on identifying particular WTD-specific "signature" changes. More information is needed on sequencing depth and whether these changes could be identified as single nucleotide variants (SNVs) in human sequences independent of contact/interaction with WTD.

Reviewer #4:

Remarks to the Author:

Thank you for inviting me to review this manuscript. The authors present a large scale surveillance study of SARS-CoV-2 (SC2) infections in free-ranging white-tailed deer (WTD) from Nov 2021 to Apr 22. In particular, they identified (1) multiple lineages of human variants of concern in white-tailed deer fuelled by frequent human-WTD transmission events, with (2) evidence of subsequent deer-to-deer transmission. They additionally identified (3) repeated substitutions of SC2 in WTD that are rare or not found in humans, with evidence of positive selection for some of them. They additionally found (4) at least 3 human-WTD-human transmission events. While this paper has the potential to be improved, most of their findings have been demonstrated previously in the literature and provides minimal advance. In particular, (1) and (2) has been discussed by at least five studies (<https://doi.org/10.1038/s41586-021-04353-x>, <https://doi.org/10.1073/pnas.2215067120>, <https://doi.org/10.1073/pnas.2121644119>, <https://doi.org/10.1038/s41467-022-30698-6>, <https://doi.org/10.1038/s41564-022-01268-9>). Additionally, (3) has been discussed by at least two studies (<https://doi.org/10.1038/s41564-022-01268-9>, <https://doi.org/10.1038/s41467-022-30698-6>) and (4) by at least one study (<https://doi.org/10.1038/s41564-022-01268-9>). However, the dataset provided in this study is large and rich, and has the potential to be analysed further to provide insights not already in the literature. My comments below are listed sequentially by line number:

Line 31: The use of the term 'bat-origin zoonotic virus' may mislead the reader to think that there was direct transmission from bats to humans as with viruses like Marburg virus, but it remains unclear how SARS-CoV-2 emerged in humans so this should be rephrased.

In-text citation are not consistently formatted – some are preceded by a space while others are not.

Line 39: Given the broad readership of the journal, it would be appropriate to more explicitly define the terms and processes here, including the concepts of animal reservoirs, human-to-animal, animal-human and human-animal-human transmission.

Lines 53-55: While may be true, it may be worth mentioning that lineages that are nearly extinct in humans were found circulating in WTD. See (<https://www.pnas.org/doi/abs/10.1073/pnas.2215067120>)

Line 71: It is not clear from the methods or results what the authors define as 'high quality RNA'. This should be explicitly stated, and how this may bias the sequence analyses performed should be discussed. It is likely that the samples that could be successfully sequenced are from deer infected at high viral loads, but would possibly miss deer with abortive infections or re-infections. If the latter is immunologically impossible, this could also be discussed.

Lines 72: It is important to distinguish between breadth of coverage and depth of coverage. For the reader to better evaluate the quality of said genomes, both are important and summaries of these metrics should be provided. Another metric that would be useful is how evenly the sequencing reads are distributed across the assembled genomes.

Fig. 1: This manuscript discusses human-to-WTD transmission, and so this figure should also include the a comparison with the geographical and temporal distribution of human SC2 samples. The trends described in lines 81-82 and 85-87, in particular, would be more easily visualised with such comparisons.

Lines 90-99: It is not clear why a the sequence identity threshold to define clusters was chosen to be 99.85%. Additionally, choosing a higher sequence identity may be problematic because the phylogenetic resolution of whom-infected-whom may be blurred if the number of substitutions between two sequences is small. This threshold should be varied and the variance of the number of clusters estimated should be provided. Additionally, confidence intervals (i.e., by bootstrapping or other means) of the number of clusters estimated should be provided. It is not clear from the methods if any masking was performed on the sequence alignments. It is well-known that multiple positions on the genome like the start or end of the SC2 genome are prone to sequencing errors and should be masked (see https://github.com/W-L/ProblematicSites_SARS-CoV2 for a list of sites). Separately, human sequences should be filtered to include only samples collected within the same time period (for e.g. (+- 1 month from collection date of deer sample). Finally, the estimates provided should be compared to those that would be obtained using a transmission-chain reconstruction approach (e.g. <https://github.com/xavierdidelot/TransPhylo> or <https://search.r-project.org/CRAN/refmans/adegenet/html/seqTrack.html>).

Lines 109-111: It would be nice to demonstrate this visually using a human+deer tree colored by county.

Lines 119-128: Implicit in this analysis is that repeated substitutions are adaptive, which should be made explicit for the reader to understand why this analysis was carried out. However, it is also important to note in the manuscript that mutations identified from these analyses are only potentially adaptive and in vitro experiments are necessary to make the claim that these mutations are indeed adaptive. The number of independent emergences for each repeated mutation should be visualised in some form and discussed. It would be reasonable to assume that mutations that emerge more often in WTD are more likely to be adaptive as opposed to being coincidental. Separately, the authors focus on positive selection but ignore negative selection or selection relaxation that are also signatures of adaptations to a novel host. See <https://github.com/veg/hyphy/issues/287> for potential approaches to investigate this. Further, it is not clear how these patterns of selection differ from those in humans. A comparison is necessary to tease out potential adaptive mutations that are host-specific. Finally, the authors touch on the potential risks of WTD-human transmission in the manuscript, but do not investigate if the potentially deer-adaptive mutations could increase immune evasion or pathogenicity in humans. Since the antibody, T-cell and B-cell epitopes of SARS-CoV-2 in humans are well-characterised, the former could be investigated by determining if some of the potentially adaptive mutations identified fall within known SC2 epitopes in databases such as IEDB (<https://www.iedb.org/>).

Lines 147-173: Local deer-to-deer transmission has been documented in multiple studies and this section does not provide new insight. There may be a potential to improve this section by describing the temporal patterns of seropositivity (prior infections) vs. PCR-positivity (active infections) during the study duration. Additionally, it remains unknown for how long a SC2 lineage can be sustained in deer populations, but this is crucial for understanding to what extent WTD can serve as a long-term

wild reservoir for the virus. The data presented in this study may be analysed to provide insights on this. For example, the time span between the oldest and newest WTD isolate within each cluster can provide a rough and likely conservative estimate of the total duration of outbreaks. Population dynamic analyses to estimate the change in viral population size over time may also shed light on this. Lines 218-219: This reminds me that the health status/symptoms of deer sampled in this study should be described to help the reader understand if a majority of SC2 infections in WTD are asymptomatic or otherwise.

Lines 218-228: This paragraph touches on an important topic but lacks nuance and may exaggerate the risk of WTD-human transmission. Of the numerous clusters of SC2 in WTD identified, only three were potential WTD-human events, indicating that WTD-human events are relatively infrequent compared to human-WTD events. Additionally, Fig. 5a shows that following WTD-human transmission, only one human isolate carrying deer-specific substitutions was identified, indicating that onward transmission of deer-specific lineages in humans is limited. There is therefore no evidence that WTD-adapted SC2 can outcompete SC2 lineages already circulating in humans. The authors should make it clear that while there is a non-negligible risk of deer-adapted lineages with increased transmissibility or pathogenicity in humans spilling back into humans, this has not been observed so far. The authors could consider merging this section with lines 261-274, which touch on a similar topic.

Responses to REVIEWER COMMENTS

Reviewer #1 (Remarks to the Author):

Overall the authors collected a significant amount of samples of SARS-CoV-2 in white tailed deer across a large geographic range. Using this data their phylogenetic analysis find convincing evidence of many spillovers to WTD and some spillovers back to humans. This study provides important information to the state of SARS-CoV-2 in wild deer populations and a potential for public health ramifications from WTD as a SARS-CoV-2 reservoir. I do have a few suggestions for improvement and requests for clarification which I have listed below:

Response: Thank you for your constructive comments. We have incorporated your comments into this revision.

Major comments

1) Identification of spillovers was done using sequences (human and deer) from only within a state. It is unclear that it is justified to partition data in this way. In particular this criteria could exclude the genetically closest human virus thus undermining the within WTD analyses, and limit the potential spillovers detected.

Response: Thank you for your comment. We acknowledge that our previous analysis may have missed the genetically closest human virus in different states when considering traveling individuals and that the limited number of human samples from some states can limit our analyses. To address this, we have updated our methods by using the UCSC UShER (Ultrafast Sample placement on Existing tRee) software (cited below) with similar sequence filtering rules as previously described (see updated online methods) to validate our findings. UShER implements the Fitch-Sankoff algorithm to infer the placement of mutations on a given tree and on the variant list and mutation-annotated tree (Turakhia et al. 2021). UShER is highly effective in handling SARS-CoV-2 data and can handle 14.3 M genomic data (up to the date March 30, 2023).

In our original analysis, we identified a human precursor virus for 264 out of 282 WTD sequences from within the same state using a sequence identity cutoff of 99.85%. We did not find genetically close human SARS-CoV-2 sequences for the remaining 18. Using UShER and the same sequence identity cutoff of 99.85%, we constructed subtrees with the top 50 closest sequences and identified a genetically close virus from humans from the same state for 234 out of the 264 WTD sequences and identified a genetically close human SARS-CoV-2 virus from a different state for the remaining 30 sequences. Additionally, UShER identified genetically close ($\geq 99.85\%$) human SARS-CoV-2 viruses from a different state for all 18 sequences that did not match in our original analysis. We then combined these out-of-state human SARS-CoV-2 sequences from UShER and those within-state sequences from our previous FastTree and CCV analyses into the phylogenetic analyses in the subsequent spillover event analyses. These analyses confirmed the 106 spillover cases identified in our earlier analyses and identified three new spillover cases involving out-of-state human sequences.

We have updated Figure 2 to clarify these spillovers as potential within-state spillover cases and have added three additional human spillover cases involving a human SARS-CoV-2 from a different state to Figure S2. In addition, we have updated both the results (Line 104-131) and methods sections (Line 423-436) to reflect these changes.

Turakhia Y, Thornlow B, Hinrichs AS, De Maio N, Gozashti L, Lanfear R, Haussler D, Corbett-Detig R. Ultrafast Sample placement on Existing tRees (USHER) enables real-time phylogenetics for the SARS-CoV-2 pandemic. *Nat Genet.* 2021 Jun;53(6):809-816. doi: 10.1038/s41588-021-00862-7. Epub 2021 May 10. PMID: 33972780; PMCID: PMC9248294.

2) *Of the sites with positive selection associated with transmission in WTD, do these overlap with sites that are under positive selection associated with transmission in humans, and other animals (ex. mink)? Interpretation of this overlap (or lack of overlap) could be informative.*

Response: Thank you for your excellent question. Several studies have investigated positive selection during SARS-CoV-2 transmission in humans and between humans and animals.

We focused on three studies that analyzed positive selection in SARS-CoV-2 viruses in humans: Rochman et al. (2021) looked at viruses before January 2021, Emam et al. (2021) focused on genes ORF1ab, S, and E, and González-Vázquez et al. (2023) reviewed positively selected substitutions during the COVID-19 pandemic. However, none of the sites they identified overlapped with the ones we identified in white-tailed deer.

In mink, four substitutions (L452M, Y453F, F486L, N501T) were reported in the spike protein during a SARS-CoV-2 outbreak on a Dutch farm (Lu et al. 2021), and three of these sites (Y453F, F486L, N501T) were observed repeatedly in mink outbreaks (Tai et al. 2022). However, these sites were also not among the ones we identified.

Bashor et al. (2021) identified 7 SNVs (H69R, D215H, D215N, N501T, D614G, H655Y, and S686G in Spike, and L37F in ORF1a) that have been reported as variants of concern in humans or other species such as cats, dogs, hamsters, and ferrets. Adaptive mutations were seen in these variants across multiple hosts. However, none of these sites were among the ones we identified in white-tailed deer.

In summary, the sites with positive selection associated with transmission that we identified in white-tailed deer were not reported in humans or other animals. This suggests that these sites identified in deer may be unique to this host or may have arisen in the deer population as a result of selective pressure on the virus.

We have added a discussion of these positive selection sites to the discussion section of the manuscript in Line 268-283.

Rochman ND, Wolf YI, Faure G, Mutz P, Zhang F, Koonin EV. Ongoing global and regional adaptive evolution of SARS-CoV-2. *Proc Natl Acad Sci U S A.* 2021 Jul 20;118(29):e2104241118. doi: 10.1073/pnas.2104241118. Epub 2021 Jul 2. PMID: 34292871; PMCID: PMC8307621.

Emam, M., Oweda, M., Antunes, A., & El-Hadidi, M. (2021). Positive selection as a key player for SARS-CoV-2 pathogenicity: Insights into ORF1ab, S and E genes. *Virus research*, 302, 198472.

González-Vázquez, L. D., & Arenas, M. (2023). Molecular Evolution of SARS-CoV-2 during the COVID-19 Pandemic. *Genes*, 14(2), 407.

Lu L, Sikkema RS, Velkers FC, Nieuwenhuijse DF, Fischer EAJ, Meijer PA, Bouwmeester-Vincken N, Rietveld A, Wegdam-Blans MCA, Tolsma P, Koppelman M, Smit LAM, Hakze-van der Honing RW, van der Poel WHM, van der Spek AN, Spierenburg MAH, Molenaar RJ, Rond J, Augustijn M, Woolhouse M, Stegeman JA, Lycett S, Oude Munnink BB, Koopmans MPG. Adaptation, spread and transmission of SARS-CoV-2 in farmed minks and associated humans in the Netherlands. *Nat Commun.* 2021 Nov 23;12(1):6802. doi: 10.1038/s41467-021-27096-9. PMID: 34815406; PMCID: PMC8611045.

Tai JH, Sun HY, Tseng YC, Li G, Chang SY, Yeh SH, Chen PJ, Chaw SM, Wang HY. Contrasting Patterns in the Early Stage of SARS-CoV-2 Evolution between Humans and Minks. *Mol Biol Evol.* 2022 Sep 1;39(9):msac156. doi: 10.1093/molbev/msac156. PMID: 35934827; PMCID: PMC9384665.

Bashor L, Gagne RB, Bosco-Lauth AM, Bowen RA, Stenglein M, VandeWoude S. SARS-CoV-2 evolution in animals suggests mechanisms for rapid variant selection. *Proc Natl Acad Sci U S A.* 2021 Nov 2;118(44):e2105253118. doi: 10.1073/pnas.2105253118. PMID: 34716263; PMCID: PMC8612357.

3) *Fig. E3 seems to indicate some reversions of the repeated amino acid mutations of interest, it would be beneficial to discuss this behavior, perhaps clarifying the patterns with mutations mapped onto a maximum likelihood tree.*

Response: Thank you for your comment. The updated Figure E2 illustrates a potential transmission event of WTD-adapted AY.103 SARS-CoV-2 from WTD to humans in North Carolina. Two WTD sequences were identified to be related to the human SARS-CoV-2 sequence NC-CORVASEQ-1086-651_2021-12-23. One of these sequences had all three repeated amino acid substitutions (ORF1a:S443P, ORF1b:V12271L, and ORF1a:T1678A) that were identical to the human SARS-CoV-2 sequence, while the other sequence (22-002350-031_NC_Randolph_2021-11-20_AY.103) was missing the ORF1b:V12271L substitution. There are two possibilities for why this substitution was missing in the second sequence: 1) there is genomic diversity among WTD SARS-CoV-2 sequences with single nucleotide variations (iSNVs); or 2) the substitution reversed back in some animals during transmission. To investigate the possibility of reverse transmission or adaptive evolution in the WTD samples, we analyzed the iSNVs among the 12 samples, defining a minor iSNV as having at least a 5% prevalence among the reads. Only one iSNV was identified in 22-002350-031_NC_Randolph_2021-11-20_AY.103 for ORF1b:V12271L, with 461 reads coding for amino acid L (39.6%) and 703 reads coding for V (60.4%). This confirms the occurrence of adaptive evolution in the WTD population. On the other hand, phylogenetic analyses suggested that these two WTD sequences associated with the human sequence formed a separate sub-lineage from the other 10 WTD sequences, lacking two of the five amino acid substitutions observed in the other WTD samples. Therefore, the likelihood of reverse transmission is low. We added the details of this iSNV to the legend of the updated Figure E2.

Minor comments

1) *In the results, line 155-159 appears to be referring to sVN which is introduced at line 161. This should be clarified.*

Response: We have moved the introduction of sVNT to Line 171 for clarity.

2) *Line 156 states the number of animals that were seropositive, what percentage of tests is this?*

Response: We have added the percentage of seropositive animals, 36.12% (332 out of 919 serum samples), to Line 172.

3) *The last 2 paragraphs of the results are somewhat difficult to follow. It was hard for me as a reader to keep track of the different clusters, perhaps name them clusters 1, 2, 3. It was also unclear based on the text, if the identified human samples were genetically linked or not (beyond the listed amino acids).*

Response: We have revised this section to improve clarity and provide a clearer illustration of the genetic relationship between human and WTD SARS-CoV-2 sequences. We updated the spillover event identifiers labeling in Figure 3b, as well as in Table S2, Figure S2 and Table S3 to reflect these changes. We also included updated information throughout the main text to reflect the changes made.

4) Line 204, the phrasing of “SARS-CoV-2 is enzootic in free-ranging WTD across nearly half of the states” seems imply that it is ruled out as being enzootic in other states, which is not the case, so I would suggest rephrasing.

Response: We have rephrased the sentence to: “SARS-CoV-2 has been detected as enzootic in free-ranging WTD across nearly half of the states” for clarity (Line 230).

5) Line 218-219, (this is likely beyond the scope of the study but would be very important to consider) is it possible to report on the symptoms of the deer, in particular if there were differences in positive and negative individuals.

Response: Thank you for this important point. The deer samples were opportunistically collected by hunters and USDA agents. None of these animals were reported with any clinical signs. This is clarified in the methods section of the revised manuscript, Line 345.

6) Line 230 appears to be missing a word

Response: We have edited this sentence for clarity by adding “compared with those from humans”. (Line 244)

7) Line 404: were sites only labeled as WTD adaptive if they were repeated and under positive selection? If this is indeed the case were sites under positive selection alone considered?

Response: We have corrected the sentence as “If a repeated amino acid substitution was under positive selection, it was defined as a WTD-adaptive substitution” to clarify that only repeated amino acid substitutions under positive selection was determined as WTD adaptive (Line 480-481).

8) In figure 1d, when was the B.1 sample collected?

Response: The B.1 sample was collected on December 01, 2021 from Pennsylvania. This has been added to the legend of Figure 1.

9) All trees should have a scale indicator

Response: We have added a scale bar to each tree and updated the figure legends to clarify that the timescale of the phylogenetic tree is represented in units of years, and the scale bar indicates the divergence time in years.

Reviewer #2 (Remarks to the Author):

The manuscript reports interesting and important results supporting previous suggestions of WTD as reservoirs for SARS-CoV-2 variants of concern, with a large number of samples and providing evidences of spillover from human to deer and spillback from deer to human. The number of tested samples collected throughout the U.S, the number of genomes retrieved and the analyses providing further evidence of spillback to humans are differentials to the established literature.

Lines 44-45: The article "From Deer-to-Deer: SARS-CoV-2 is efficiently transmitted and presents broad tissue tropism and replication sites in white-tailed deer" (Martins et al., 2022) should also be cited here. It is one of the most important on the infection and transmission dynamics of SARS-CoV-2 in WTD.

Response: Thank you for your comment. This is an important article and we have cited this article in Line 49.

It is not clear in the methods or results if the samples used in this study were selected among a larger number of samples and how or if these states were selected for sampling.

Response: Only 27 states participated in the surveillance conducted in this study, and we have included all available data in our analysis. Of the 9,091 collected WTD samples, 1,116 were SARS-CoV-2-positive, and only 346 had sufficient genomic coverage for further analyses. Of these, high sequence coverage (>95%) was obtained for 282 samples. We included genomic data from all 282 remaining in our analyses and did not exclude any samples. We clarified this at Line 72 and Line 345.

Extended Data Figure E6: The phylogenetic tree does not contain all the WTD sequences from public databases. Otherwise, it should be mentioned how these 118 WTD sequences were selected to build this tree.

Response: We added clarification that all available WTD SARS-CoV-2 genomes (332 sequences) as of December 5, 2022 were downloaded from GISAID. Of these, 118 contained complete and high coverage sequences (> 99% coverage). These 118 complete sequences were selected for phylogeographic analyses. We clarified this detail in Line 503-506 and the figure legend of the updated Fig. E5.

Reviewer #3 (Remarks to the Author):

Numerous groups have demonstrated that white-tailed deer (WTD) are susceptible to SARS-CoV-2 and may be a source for zoonotic infection of humans. The data provided in this work supports these prior studies and goes on to suggest that SARS-CoV-2 lineages can not only be introduced but can become enzootic and co-circulate in WTD. Specifically, the authors use a combination of sequence analysis and sero-epidemiology to demonstrate transmission within WTD populations. This data is compelling and will be important to monitor to better understand viral evolution within WTD populations within the US.

My primary concerns relate to the data suggesting reverse zoonosis infection of WTD-adapted viruses to humans. Specific questions:

1. Without corresponding human sequences from the same geographic location of the deer, how can the authors "prove" that these changes weren't common in humans and spilling over into WTD?

Response: Thank you for your comment. We indeed acknowledge the possibility that some repeated amino acid substitutions could arise from transmission events in humans and then be transmitted back to white-tailed deer (WTD). To address this concern, we focused only on amino acid substitutions that were identified in at least two independent transmission events. Additionally, we conducted a thorough check of all available human SARS-CoV-2 data and confirmed that these amino acid substitutions are infrequent in human epidemic viruses. This approach increased our confidence that the WTD-specific amino acid substitutions we identified were not simply the result of transmission from humans. We have expanded our discussion section to clarify the limitations of our study.

2. The WTD-human conclusions are based on identifying particular WTD-specific "signature" changes. More information is needed on sequencing depth and whether these changes could be identified as single nucleotide variants (SNVs) in human sequences independent of contact/interaction with WTD.

Response: Thank you for your response. We have addressed your concerns by updating Table S1 with sequence coverage, average reading depth, and the total number of sites with at least 10 reads. While we acknowledge that the observed amino acid substitutions may have resulted from switches in the proportions of single nucleotide variants (SNVs), we were unable to conduct SNV analyses using public human SARS-CoV-2 data due to the lack of necessary next generation sequencing data. Instead, we analyzed 148 human SARS-CoV-2 viruses from Missouri that belong to the same lineages as AY.103, AY.119, and AY.44 with our Human-WTD-human events. We used the CDC IRMA (Iterative Refinement Meta-Assembler) pipeline with a Phred quality score of 20 trimmed for the single nucleotide variants analysis and defined minority SNVs as those with at least 0.05% of reads but less than 50%, as shown in Table S7 we added into this resubmission. Our analysis confirmed that we did not observe any SNVs at the sites where we detected WTD-specific repeated amino acid substitutions in the Human-WTD-human events, as stated in Line 219-224. The results are also included in Table S8.

Reviewer #4 (Remarks to the Author):

Thank you for inviting me to review this manuscript. The authors present a large scale surveillance study of SARS-CoV-2 (SC2) infections in free-ranging white-tailed deer (WTD) from Nov 2021 to Apr 22. In particular, they identified (1) multiple lineages of human variants of concern in white-tailed deer fuelled by frequent human-WTD transmission events, with (2) evidence of subsequent deer-to-deer transmission. They additionally identified (3) repeated substitutions of SC2 in WTD that are rare or not found in humans, with evidence of positive selection for some of them. They additionally found (4) at least 3 human-WTD-

human transmission events. While this paper has the potential to be improved, most of their findings have been demonstrated previously in the literature and provides minimal advance. In particular, (1) and (2) has been discussed by at least five studies (<https://doi.org/10.1038/s41586-021-04353-x>, <https://doi.org/10.1073/pnas.2215067120>, <https://doi.org/10.1073/pnas.2121644119>, <https://doi.org/10.1038/s41467-022-30698-6>, <https://doi.org/10.1038/s41564-022-01268-9>). Additionally, (3) has been discussed by at least two studies (<https://doi.org/10.1038/s41564-022-01268-9>, <https://doi.org/10.1038/s41467-022-30698-6>) and (4) by at least one study (<https://doi.org/10.1038/s41564-022-01268-9>). However, the dataset provided in this study is large and rich, and has the potential to be analysed further to provide insights not already in the literature.

Response: Thank you for your comments and for listing those recent studies. We have indeed acknowledged these important studies and discussed their findings with the context of our manuscript.

My comments below are listed sequentially by line number:

Line 31: The use of the term ‘bat-origin zoonotic virus’ may mislead the reader to think that there was direct transmission from bats to humans as with viruses like Marburg virus, but it remains unclear how SARS-CoV-2 emerged in humans so this should be rephrased.

Response: We have revised it as “a zoonotic virus” for clarity (Line 31).

In-text citation are not consistently formatted – some are preceded by a space while others are not.

Response: We have updated the in-text citations for consistency.

Line 39: Given the broad readership of the journal, it would be appropriate to more explicitly define the terms and processes here, including the concepts of animal reservoirs, human-to-animal, animal-human and human-animal-human transmission.

Response: We have updated this section with additional definitions.

“An animal reservoir for SARS-CoV-2 refers to a host in which the virus circulates covertly, persisting in the population and can be transmitted to other animals or humans potentially causing disease outbreaks.” (Line 39-41)

Lines 53-55: While may be true, it may be worth mentioning that lineages that are nearly extinct in humans were found circulating in WTD. See (<https://www.pnas.org/doi/abs/10.1073/pnas.2215067120>)

Response: We agree that it is important to note that circulating lineages in WTD were already extinct in humans, and we did include a discussion on the specific lineages found in deer that are nearly extinct in humans. The reference you suggested has already been cited.

“In another study, Caserta et al. detected Alpha and Gamma viruses in New York WTD between October and December 2021 after these VOCs were replaced by Delta and Omicron in humans, and detected Delta viruses in New York WTD between October and November 2021 during the period when Delta and Omicron were predominant in humans¹⁸.” (Line 236-239)

Line 71: It is not clear from the methods or results what the authors define as ‘high quality RNA’. This should be explicitly stated, and how this may bias the sequence analyses performed should be discussed. It is likely that the samples that could be successfully sequenced are from deer infected at high viral loads, but would possibly miss deer with abortive infections or re-infections. If the latter is immunologically impossible, this could also be discussed.

Response: Thank you for making this important observation, which is a common challenge for all genomic data for infectious diseases. When viral loads are too low, it can be impossible to recover full genomic sequences. In our study, we defined high quality RNA which leads to 95% or above full coverage of the viral genome. Please see our method section:

“Among these samples, only those with a high sequencing coverage over >95% of the reference genome (n=282), an IRMA score of 95%, were selected for further evolutionary analyses.” (Line 383-384)

Lines 72: It is important to distinguish between breadth of coverage and depth of coverage. For the reader to better evaluate the quality of said genomes, both are important and summaries of these metrics should be provided. Another metric that would be useful is how evenly the sequencing reads are distributed across the assembled genomes.

Response: Thank you for bringing this to our attention. We have recently updated Table S1 to include additional metrics, such as the genomic size, average depth of coverage, and total number of sites with at least 10 reads. Although uniform coverage depth across the genome is uncommon, we aimed to provide a comprehensive overview of the coverage depth in our study.

Fig. 1: This manuscript discusses human-to-WTD transmission, and so this figure should also include the a comparison with the geographical and temporal distribution of human SC2 samples. The trends described in lines 81-82 and 85-87, in particular, would be more easily visualised with such comparisons.

Response: Thank you, we generated a supplementary figure illustrating all 39 cases of subsequent local deer-to-deer transmission. Indeed, this map enables more clear visualization of local transmission. We added Figure S3 in Line 118.

Lines 90-99: It is not clear why a the sequence identity threshold to define clusters was chosen to be 99.85%. Additionally, choosing a higher sequence identity may be problematic because the phylogenetic resolution of whom-infected-whom may be blurred if the number of substitutions between two sequences is small. This threshold should be varied and the variance of the number of clusters estimated should be provided. Additionally, confidence intervals (i.e., by bootstrapping or other means) of the number of clusters estimated should be provided.

Response: Thank you for your critical comment. In our study, we used all available human SARS-CoV-2 sequences downloaded from public databases (i.e., GISAID and NCBI) to identify the potential precursor virus for the WTD SARS-CoV-2 viruses. We employed both sequence identity (with a cutoff of 98.85%) and the posterior probabilities of the genetic clusters to determine whether a human SARS-CoV-2 genome was a precursor virus using time-scaled Bayesian phylogenetic analyses (Table S2). The posterior probabilities and the confidence intervals are shown in Supplementary Figure S2. We would like to clarify that although 98.85% was chosen as an arbitrary value, some human SARS-CoV-2 sequences have more than 98.85% genomic sequence identity to a WTD SARS-CoV-2 sequence. Nonetheless, our goal in identifying the precursor virus was to identify independent transmission events, and we do not claim that the virus was transmitted directly from the individual from whom we obtained the sequence.

It is not clear from the methods if any masking was performed on the sequence alignments. It is well-known that multiple positions on the genome like the start or end of the SC2 genome are prone to sequencing errors and should be masked (see https://github.com/W-L/ProblematicSites_SARS-CoV2 for a list of sites).

Response: While we did not perform any masking on the sequence alignments in order to preserve the completeness of the sequencing. But the 5' untranslated region (before position 266) and 3' untranslated region (after position 29,674) of the genome were not included in the nucleotide and amino acid substitution analyses because they are not in the amino acid coding regions.

We also compared the positions containing mutations identified in our study to the regions listed in the linked forum and did not find any mutations among positions that are prone to sequencing errors. However, it is worth noting that the data used in the previous analyses were from human SARS-CoV-2 genomes, and we did not have as large of a sample size for the animal samples. Therefore, sites with sequencing errors in the animal samples may require further exploration. This is clarified in the revised manuscript.

“Because of low coverage at the 5' untranslated region (before position 266) and 3' untranslated region (after position 29,674) of the genome, we excluded these positions from amino acid substitution analyses.” (Line 376-378)

Separately, human sequences should be filtered to include only samples collected within the same time period (for e.g. (+- 1 month from collection date of deer sample).

Response: Thank you for your suggestion. Initially, we attempted to focus on human sequences collected during a short time window that aligned with the dates when the WTD sequences were collected. However, we found that the number of samples was too small, particularly for those from the same state, which would limit our analyses after the viruses circulated in WTD but before their precursor viruses became subdominant. As a result, we decided to include all available samples in our analyses.

Finally, the estimates provided should be compared to those that would be obtained using a transmission-chain reconstruction approach (e.g. <https://github.com/xavierdidelot/TransPhylo> or <https://search.r-project.org/CRAN/refmans/adegenet/html/seqTrack.html>).

Response: Thank you for recommending TransPhylo as a useful tool for reconstructing transmission chains. We used TransPhylo to analyze the transmission events in the B.1.1.7 branch, which had the largest number of WTD samples and the most detailed transmission events in our phylogeographic analyses (see RL.Figure 1). The spillover events from humans to WTD and from WTD to WTD were consistent with those in the Human-WTD clusters and Human-WTD-WTD clusters from our previous analyses. However, TransPhylo claimed additional transmission events in some branches with low posterior probability, as it assumes all tree topologies have high confidence. This is not always the case, as many of these SARS-CoV-2 sequences have high sequence identities and some branches have low posterior probability and thus lack sufficient resolution. While the results from TransPhylo support our hypothesis that SARS-CoV-2 was transmitted in the WTD population within and across counties in the state of New York, we decided not to include these results in the updated manuscript due to the potential low confidence in some transmission events referred by TransPhylo.

Given that BEAST is one of the most widely used tools in phylogeographic analyses, we have decided to present only the results from our BEAST analyses in the revised manuscript. We have made several updates to both our results and methods, and we have also added Bayes Factors to Table S5 to provide additional support for our findings. We believe that these revisions have clarified these sections of the manuscript.

RL.Figure 1. Reconstruction of transmission chain for SARS-CoV-2 viruses in the New York WTD population. We reconstructed the transmission chain for SARS-CoV-2 viruses in the WTD population of New York using TransPhylo to analyze the transmission events, including the B.1.1.7 branch with scale and shape of Gamma model estimated by Bayesian phylogenetic analyses. Human sequences are indicated in blue, while WTD sequences are indicated in maroon. Each horizontal line represents a case, with the darkness of the line indicating the estimated changing infectivity over time. Vertical arrows represent transmission from one case to another, while red circles indicate the individuals that were sampled and when they were sampled.

Lines 109-111: It would be nice to demonstrate this visually using a human+deer tree colored by county.

Response: Thank you for your feedback. We have attempted to color different counties in the trees but found that it made the figure convoluted and difficult to interpret. Therefore, we have created a new supplementary figure, Figure S3, which clearly illustrates the local deer-to-deer transmission within and across different counties. We have also updated the text in Line 118 to reflect the addition of this supplementary figure.

Lines 119-128: Implicit in this analysis is that repeated substitutions are adaptive, which should be made explicit for the reader to understand why this analysis was carried out. However, it is also important to note in the manuscript that mutations identified from these analyses are only potentially adaptive and in vitro experiments are necessary to make the claim that these mutations are indeed adaptive.

Response: Thank you for your comment. We have clarified the concept of repeated substitutions in the method section “If a repeated amino acid substitution was under positive selection, it was defined as a WTD adaptive substitution”. (Line 480-481). Furthermore, we have added a discussion on the importance of additional experiments for validating our findings in the limitations section “The roles of these reoccurring substitutions in WTD adaptation need further investigation”. (Line 265-266)

The number of independent emergences for each repeated mutation should be visualised in some form and discussed. It would be reasonable to assume that mutations that emerge more often in WTD are more likely to be adaptive as opposed to being coincidental.

Response: To investigate the independent emergence of repeated amino acid substitutions, we added the spillover event identifiers in the updated Figure 3b, which can be referred in Figure S2 and Table S2. The results show clearly that these substitutions emerged at least twice from separate genetic lineages across the tree, supporting their adaptive nature. Thus, these results suggest that these amino acid substitutions are not random but likely to be adaptive.

Separately, the authors focus on positive selection but ignore negative selection or selection relaxation that are also signatures of adaptations to a novel host. See <https://github.com/veg/hyphy/issues/287> for potential approaches to investigate this.

Response: We agree that both positive selection and negative selection are important signatures of adaptations to a novel host. Among the 112 amino acid substitutions detected in at least two independent transmission events, only one was determined to be under negative selection when analyzed using FUBAR. We have added the result from the negative selection analysis to Supplementary Table S3. The methods are further updated (Line 464-471).

Further, it is not clear how these patterns of selection differ from those in humans. A comparison is necessary to tease out potential adaptive mutations that are host-specific.

Response: We have added a comparison to human patterns of selection in the discussion (Line 281-282). Please also see the response to the second comment from Reviewer #1.

Finally, the authors touch on the potential risks of WTD-human transmission in the manuscript, but do not investigate if the potentially deer-adaptive mutations could increase immune evasion or pathogenicity in humans. Since the antibody, T-cell and B-cell epitopes of SARS-CoV-2 in humans are well-characterised, the former could be investigated by determining if some of the potentially adaptive mutations identified fall within known SC2 epitopes in databases such as IEDB (<https://www.iedb.org/>).

Response: Thank you for your suggestion. As we currently lack knowledge about WTD-specific T-cell and B-cell epitopes, it would be challenging for us to infer whether the observed mutations were derived from WTD herd immunity. However, we analyzed whether the deer-adaptive mutations were located in human-specific T-cell and B-cell epitopes of SARS-CoV-2, as these are well-studied and could potentially provide insights into the adaptive significance of the mutations. Our results showed that 51 out of 58 deer-adaptive amino acid substitutions are located in those reported human T-cell (n=9) or B-cell epitopes (n=42) of SARS-CoV-2 (from the Immune Epitope Database), and the details are listed in Table S9. We have added related text at Line 269-275.

Lines 147-173: Local deer-to-deer transmission has been documented in multiple studies and this section does not provide new insight. There may be a potential to improve this section by describing the temporal patterns of seropositivity (prior infections) vs. PCR-positivity (active infections) during the study duration. Additionally, it remains unknown for how long a SC2 lineage can be sustained in deer populations, but this is crucial for understanding to what extent WTD can serve as a long-term wild reservoir for the virus. The data presented in this study may be analysed to provide insights on this. For example, the time span between the oldest and newest WTD isolate within each cluster can provide a rough and likely conservative estimate of the total duration of outbreaks. Population dynamic analyses to estimate the change in viral population size over time may also shed light on this.

Response: Thank you for your important comment. The deer samples used in our study were opportunistically collected by hunters and USDA agents. Unfortunately, the number of samples collected from the same sampling sites was limited, typically spanning less than two weeks, which prevented us from studying virus population dynamics in WTD. We have added a discussion on this limitation in our manuscript, Line 327-332.

Lines 218-219: This reminds me that the health status/symptoms of deer sampled in this study should be described to help the reader understand if a majority of SC2 infections in WTD are asymptomatic or otherwise.

Response: None of these animals were reported with any clinical signs. This is clarified in the revised manuscript, Line 345-346.

Lines 218-228: This paragraph touches on an important topic but lacks nuance and may exaggerate the risk of WTD-human transmission. Of the numerous clusters of SC2 in WTD identified, only three were potential WTD-human events, indicating that WTD-human events are relatively infrequent compared to human-WTD events. Additionally, Fig. 5a shows that following WTD-human transmission, only one human isolate carrying deer-specific substitutions was identified, indicating that onward transmission of deer-specific lineages in humans is limited. There is therefore no evidence that WTD-adapted SC2 can outcompete SC2 lineages already circulating in humans. The authors should make it clear that while there is a non-negligible risk of deer-adapted lineages with increased transmissibility or pathogenicity in humans spilling back into humans, this has not been observed so far.

Response: Thank you for your comment and for acknowledging the limited WTD-human transmission observed in our study, which contrasts with the human-to-WTD transmission events we observed. While we recognize the relatively low risk of WTD-to-human transmission, we believe it is important to report our findings as this is the first documented report of WTD-to-human transmission in the United States. However, we caution that our sample size is limited, and further studies are needed to fully understand the extent of WTD-to-human transmission and its potential public health implications. We have updated the discussion to ensure that our results are not overstated:

“Our findings highlight the potential public health implications of WTD-to-human transmission, but further studies with larger sample sizes are needed to fully understand the extent of transmission and the associated risks to human health.” Line 296-298.

The authors could consider merging this section with lines 261-274, which touch on a similar topic.

Response: Thank you for your feedback! We have moved Section 218 down to combine two sections about transmission.

Reviewers' Comments:

Reviewer #1:

Remarks to the Author:

I have reviewed the authors' responses and feel that the authors have addressed my concerns.

Reviewer #2:

Remarks to the Author:

The author's responses were clarifying and satisfactory. The changes to the manuscript have improved its quality.

Reviewer #3:

Remarks to the Author:

The authors were responsive to previous reviewer's concerns. No further comments.

Reviewer #4:

Remarks to the Author:

The manuscript has significantly improved following revision. Some outstanding comments:

1. The motivation for the T/B-cell analysis needs to be explained more clearly, that is, to test whether these mutations would lead to immune escape if these WTD-associated variants spilled back into humans. The fact that so many of these putatively WTD-adaptive mutations fall within known human T-cell epitope sequences, indicates a risk for greater immune escape of WTD-adapted strains. Of course, more in vitro work needs to be conducted to confirm this, and I don't expect the authors to perform these analyses for the purposes of this paper, but these implications and caveats should be included in the discussion.
2. For the sequence masking, I do not understand what the authors mean by 'to preserve the completeness of the sequencing'. The motivation for sequence masking is to ensure that any reconstructed phylogeny is not informed by erroneous mutations in the sequence alignment. I don't see a good reason for not masking out these positions.
3. The authors have not fully addressed my previous query: 'This [percentage identity] threshold should be varied and the variance of the number of clusters estimated should be provided. Additionally, confidence intervals (i.e., by bootstrapping or other means) of the number of clusters estimated should be provided.' I maintain, that these confidence intervals are necessary for the reader to understand the uncertainty in their estimates.

Response Letter

The manuscript has significantly improved following revision. Some outstanding comments:

1. The motivation for the T/B-cell analysis needs to be explained more clearly, that is, to test whether these mutations would lead to immune escape if these WTD-associated variants spilled back into humans. The fact that so many of these putatively WTD-adaptive mutations fall within known human T-cell epitope sequences, indicates a risk for greater immune escape of WTD-adapted strains. Of course, more in vitro work needs to be conducted to confirm this, and I don't expect the authors to perform these analyses for the purposes of this paper, but these implications and caveats should be included in the discussion.

Response: Thank you! It is intriguing to observe that WTD-adaptive substitutions are found within a significant number of human T-cell epitopes. This phenomenon has the potential to diminish the impact of T-cell memory generated from prior SARS-CoV-2 infections within the human population. In the revised manuscript, we have included an additional section in the discussion to address this finding (Line 274-276).

2. For the sequence masking, I do not understand what the authors mean by 'to preserve the completeness of the sequencing'. The motivation for sequence masking is to ensure that any reconstructed phylogeny is not informed by erroneous mutations in the sequence alignment. I don't see a good reason for not masking out these positions.

Response: Thank you for your feedback. We appreciate your insights regarding the benefits of masking certain genomic regions. We acknowledge that masking can effectively address regions prone to sequencing errors, high genetic variation, biases from recombination, and positive selection. By employing this technique, we agree that we can enhance the accuracy and reliability of our analyses by reducing noise, biases, and uncertainties associated with specific genomic positions.

In response to your comments, we made a concerted effort to construct a tree focusing on the 265 problematic positions summarized at https://github.com/W-L/ProblematicSites_SARS-CoV2. Notably, the tree topologies derived from the complete genome tree and the tree marked with the reported positions exhibited striking similarity (Fig. R1). By excluding the reported positions into the tree, we performed a reanalysis of the transmission events. Encouragingly, the number of transmission events remained unchanged when using the same identification rules. Consequently, it appears that the masked positions do not significantly impact the analyses conducted in this study.

We added the mask tree as a supplementary figure (Fig. S4). In addition, we revised the online method section accordingly.

Fig. R1. Phylogenetic analyses of the WTD SARS-CoV-2 sequences ($n = 282$) and their potential precursor viruses in humans. Because of low coverage at the 5' untranslated region (before position 266) and 3' untranslated region (after position 29,674) of the genome, we excluded these positions from nucleotide and amino acid substitution analyses. In addition, 265 problematic positions summarized at https://github.com/W-L/ProblematicSites_SARS-CoV2 were marked before phylogenetic analyses. The estimates of divergence time were obtained by calculating the median node height of the 95% highest posterior density (HPD) interval from a maximum clade credibility tree generated using BEAST. The node bars, depicted in light blue, represent the 95% HPD interval for each node. The timescale of the phylogenetic tree was represented in units of years, and the scale bar indicates the divergence time in years.

3. *The authors have not fully addressed my previous query: 'This [percentage identity] threshold should be varied and the variance of the number of clusters estimated should be provided. Additionally, confidence intervals (i.e., by bootstrapping or other means) of the number of clusters estimated should be provided.' I maintain, that these confidence intervals are necessary for the reader to understand the uncertainty in their estimates.*

Response: Thank you for providing this valuable feedback. We appreciate your insights. Regarding the genomic sequences of WTD SARS-CoV-2 viruses and the potential human SARS-CoV-2 precursor viruses from public databases, we observed a substantial degree of similarity. The majority of WTD SARS-CoV-2 viruses exhibited a sequence identity exceeding 99.80% (Figure R2a).

To evaluate the robustness of the parameters used in defining transmission events (sequence identity and posterior probability), we conducted further analyses across a range of these parameters. As depicted in Fig. R2b, our results indicated that the number of transmission events remained quite consistent when employing a sequence identity cutoff of 99.85% or lower. For example, we identified a total of 110 spillover events with the sequence identity of 99.80% or less. On the other hand, when the sequence identity cutoff became more stringent, the number of clusters decreased significantly due to the absence of corresponding human SARS-CoV-2 precursor viruses. For example, we identified a total of 100 spillover events with the sequence identity of 99.90% and only 66 with the sequence identity of 99.95%.

Conversely, the posterior probability did not have a notable impact on the number of transmission events, as the tree subclades associated with the majority of these events exhibited a posterior probability of 0.9 or higher. Only the subclade associated with one human-WTD-human spillover event had a posterior probability of 0.78 and another one with human-WTD spillover event had a posterior probability of 0.66.

Overall, these additional analyses support the robustness of the criteria used in our transmission event analyses. In the revised manuscript, we have included these results in the Online Methods section and Fig. S5 to provide readers with a better understanding of how these parameters were selected.

Fig. R2. Parameter optimization for determining transmission events associated with WTD SARS-CoV-2 viruses. a) Distribution of genomic nucleotide sequence identities between each WTD SARS-CoV-2 and the human potential precursor viruses selected from public database. The number of b) the total spillover events, c) the Human-WTD spillover events, d) the Human-WTD-WTD spillover events, and e) the Human-WTD-Human spillover events were determined by altering the genomic nucleotide sequence identities varying from 99.5 to 99.95% and the posterior probability varying from 0.60 to 0.90.

Reviewers' Comments:

Reviewer #4:

Remarks to the Author:

Thanks to the authors for revising the manuscript and providing additional analyses so quickly. Overall, the work presented in this revised manuscript is robust and I recommend publication.